# X-chromosome upregulation operates on a gene-by-gene basis at RNA and protein levels

Ryan N. Allsop[1,2,7], Jeffrey Boeren[3,7], Beatrice F. Tan [3,7], Sarra Merzouk[3], Suresh Poovanthingal[2,4], Wilfred F. J. van IJcken [5], Jeroen A. A. Demmers [6], Hegias Mira Bontenbal [3], Cristina Gontan[3], Joost Gribnau [3,8] ✉ & Vincent Pasque [1,2,8] ✉

Gene dosage compensation mechanisms are crucial for mammalian development. In mice, recent findings show that cells can sense the number of X chromosomes. Loss or inactivation of one of the two X chromosomes is compensated by upregulating the remaining active X chromosome, a process termed X-chromosome upregulation (XCU). However, how cells sense X-chromosome dosage and induce XCU remains unclear. Here, we show that heterozygous X chromosome fragment deletions in mouse pluripotent stem cells induces XCU in trans, and that compensation takes place at the mRNA and protein level. Furthermore, we found that inducing gene silencing *in cis* on autosomes induces gene dosage compensation in trans. This work provides significant insights into the molecular foundations of mammalian gene dosage compensation.

Most mammals are diploid, with two copies of each chromosome per cell[1]. In some cases, however, the number of chromosomes or segments of chromosomes can change. Changes in gene copy number, also known as changes in gene dosage, have been closely linked to human diseases[2,3]. Gene dosage compensation mechanisms attempt to resolve imbalances in gene dosage[4]. However, how gene dosage compensation is triggered in mammals remains unclear.

A paradigm to study gene dosage compensation mechanisms is the X chromosome. In placental mammals, the female sex is typically characterized by two X chromosomes, while the male sex typically has one X chromosome and one Y chromosome. This imbalance in X-linked gene dosage between sexes is thought to have driven the evolution of X-linked dosage compensation mechanisms[4–11]. In female mammals, one of the two X chromosomes expresses the long non-coding RNA *Xist* and is inactivated *in cis* (on the same chromosome) during early development, a process termed X-chromosome inactivation (XCI)[12–15]. As a result, there is only one active X chromosome in female somatic cells, while male XY somatic cells also have one active X chromosome.

In contrast to the X chromosome, each autosome is present in two active copies. In 1963, Susumu Ohno hypothesized that a dosage compensation mechanism evolved to upregulate X-linked gene expression on the active X chromosome, to balance X-linked gene dosage with the gene dosage of diploid autosomes[9]. This process is referred to as X-chromosome upregulation (XCU)[4]. Recent studies using allele-resolution transcriptome analyses have confirmed that XCU takes place at the transcriptional level in mouse cells with a single active X chromosome, and that XCU is developmentally regulated[4,12,16–34].

Multiple lines of evidence using allele-resolution transcriptome analyses have indicated that mammalian cells can sense the number of X chromosomes they have, as well as the transcriptional status of the X chromosome, and compensate X-linked gene expression accordingly[16,18,20]. Before XCI, female diploid XX mouse pluripotent stem cells (PSCs) have two active X chromosomes (Xa), while male XY cells have only one. By consequence, female mouse PSCs have higher levels of X-linked gene products than male cells. However, recent studies showed that unlike previously thought, each active X

[1]Department of Development and Regeneration, University of Leuven (KU Leuven), Leuven, Belgium. [2]KU Leuven Institute for Single Cell Omics (LISCO), Leuven, Belgium. [3]Department of Developmental Biology, Erasmus University Medical Center, Rotterdam, The Netherlands. [4]VIB-KU Leuven Center for Brain & Disease Research, Leuven, Belgium. [5]Erasmus Center for Biomics, Erasmus University Medical Center, Rotterdam, the Netherlands. [6]Proteomics Center, Erasmus University Medical Center, Rotterdam, The Netherlands. [7]These authors contributed equally: Ryan N. Allsop, Jeffrey Boeren, Beatrice F. Tan. [8]These authors jointly supervised this work: Joost Gribnau, Vincent Pasque. ✉e-mail: j.gribnau@erasmusmc.nl; vincent.pasque@kuleuven.be

chromosome in female XX mouse PSCs and naive epiblast are not upregulated, consistent with no requirement for dosage compensation between two active X chromosomes and diploid autosomes[16,20]. Furthermore, when XX embryonic fibroblasts, with one upregulated active (Xu) and one inactive X chromosome (Xi), are reprogrammed to induced pluripotent stem cells (iPSCs), reactivation of the Xi occurs concomitantly with loss of upregulation from the Xu[16]. A study in mouse iPSCs showed that when one of the two non-upregulated Xa chromosomes is lost in PSCs, XCU is induced on the remaining X chromosome to compensate gene dosage imbalance with diploid autosomes[16]. This observation was later confirmed in mouse embryonic stem cells (mESCs)[20]. Similarly, XCU is also observed in male XY mESCs[20]. In summary, when both X chromosomes are active, there is no XCU. However, when only one active X chromosome is present, XCU is triggered on the remaining active X chromosome. Collectively, these results suggest that mammalian cells can sense the number of active X chromosomes and compensate for the presence of a single active X chromosome by XCU. However, how cells sense the number of active X chromosomes and trigger compensation for gene dosage imbalances, and if they can sense when specific X chromosome regions or genes are missing, remains unclear.

The molecular mechanisms controlling XCU in mammals are still unknown. Previous studies have attempted to identify active histone modifications associated with XCU in mice; however, the results have been conflicting or unclear. One study reported enrichment of histone 4 lysine 16 acetylation (H4K16ac) on the Xu, but analyzed cells with two active X chromosomes, which we now know do not undergo XCU[29]. Another study reported enrichment of histone 3 lysine 4 tri-methylation (H3K4me3) as well as histone 3 lysine 36 tri-methylation (H3K36me3) on the Xu; however, this enrichment was compared to autosomes, rather than to the Xa[25]. Finally, a recent study found no enrichment of any common histone modifications on the Xu compared to the Xa[20]. Therefore, it remains unclear whether histone modifications are enriched on the mammalian Xu.

Changes in gene dosage of autosomes is a hallmark of cancer and early human IVF embryos[35–39]. Studies in human cancer cells and human embryos have suggested that gene dosage compensation can occur on autosomes at both the transcriptional and translational levels[35,40]. However, it is unknown if these effects can also be induced by gene silencing on an autosome, as opposed to genetic loss.

In this study, we sought to investigate how the X chromosome and autosomes sense and compensate for changes in gene dosage in mammals. We use CRISPR-Cas9 to generate large heterozygous X-linked deletions in female XX mESCs, followed by allele and molecule-resolution single-cell and bulk transcriptomics, as well as quantitative proteomics to measure the impact of the deletions on gene expression. We find evidence for dosage compensation by increased transcription and protein levels specifically within the deleted regions in trans. Furthermore, we investigated whether dosage compensation of autosomal genes can occur without loss of genetic material, and found evidence for transcriptional dosage compensation in trans (on the other chromosome) when one of two alleles is inactivated in cis. In conclusion, this work establishes that mammalian cells can sense the heterozygous loss or inactivation of specific fragments of chromosomes or genes and compensate these gene dosage imbalances by upregulation at the transcriptional and protein levels in trans.

## Results

### Allele and molecule-resolution analysis of XCU in mouse PSCs

To investigate XCU, we used single-cell RNA-sequencing (scRNA-seq) in mouse PSCs with a range of genotypes. The cell lines originate from a cross between two mouse subspecies (*Mus musculus musculus* × *Mus musculus castaneus, Mus/Cast*). The cells were (1) female iPSCs with two active X chromosomes ($X^{Cast}X^{Mus}$), (2) male ESCs ($X^{Mus}Y$), and (3) female mESCs with X-chromosome monosomy of the paternal or maternal X chromosome (XO, either $X^{Cast}O^{Mus}$ or $X^{Mus}O^{Cast}$) (Fig. 1a and Supplementary Fig. 1a)[41]. Previous studies indicated that both mESCs and iPSCs behave similarly with respect to X chromosome dosage compensation properties[16,20,41–43]. We applied Smart-seq3xpress, a plate-based scRNA-seq method providing full transcript coverage with the ability to count the number of transcript molecules using unique molecular identifiers (UMIs)[44]. Transcripts were further split by allele of origin based on a list of ~17.5 million single-nucleotide polymorphisms (SNPs) unique to each subspecies[16,45,46], providing allele-specific resolution data.

To quantify gene expression from each X chromosome, we measured the mean normalized expression (transcripts per million, TPM) of all expressed X-linked genes from each allele for each cell line. This enabled us to identify which X-chromosome was lost in the XO cells (Supplementary Fig. 1b). XX iPSCs and ESCs have been previously described to lose one of the two X chromosomes[41,43,47,48]. Consistent with this, after quality filtering, we obtained 6 cells which lost the *Cast* allele ($X^{Mus}O^{Cast}$) and 44 cells which lost the *Mus* allele ($X^{Cast}O^{Mus}$). As expected, we observed significantly higher allelic expression levels when only one active X chromosome is present, in $X^{Mus}Y$ and $X^{Mus}O^{Cast}$ cells (Fig. 1b and Supplementary Fig. 2a). When the *Mus* allele rather than the *Cast* X chromosome allele was lost, in $X^{Cast}O^{Mus}$ cells, X-linked gene expression was increased on the *Cast* allele (Fig. 1c). In summary, loss of one X chromosome in XX mouse PSCs induces global compensation of gene expression on the remaining monosomic X chromosome. Allelic X-to-autosome ratio (X:A) analysis confirmed elevated X:A ratios in cells with a single X chromosome, while $X^{Cast}X^{Mus}$ cells exhibited no increased X:A ratio, confirming previous findings[16,20] (Fig. 1d, e and Supplementary Fig. 2b). These findings confirm the presence of XCU in diploid cells with a single X chromosome[16], and that mouse PSCs can sense the number of X chromosomes and adapt gene expression by undergoing XCU when only one Xa is present. Consistent with previous studies, the average expression from the *Mus* allele tended to be slightly higher than the *Cast* allele, a slight bias most likely induced to mapping reads to the *Mus musculus musculus* reference genome[16,18,49]. Together, these results confirm that transcriptional dosage compensation by XCU occurs in mouse PSCs with a single active X chromosome, but not with two active X chromosomes. Moreover, we can accurately measure and identify XCU using a hybrid/polymorphic mouse PSC system combined with allele-specific expression analysis using Smart-seq3xpress scRNA-seq.

### Identification of genes subject to XCU

We next sought to characterize which X-linked genes are transcriptionally dosage compensated in this system. Of the 405 expressed X-linked genes for which we obtained allele-resolution expression data, 103 (~40%) were found to be significantly transcriptionally upregulated in XO and XY cells (collectively $X^{monosomy}$) compared to XX cells (Supplementary Fig. 2c and Supplementary Table 1). These results are in line with previous findings that transcriptional XCU is specific to a subset of genes[17]. Both non-upregulated and upregulated genes were found to be located across the X chromosome and not enriched to a specific locus (Supplementary Fig. 2d). The mean allelic expression of the upregulated genes was significantly higher in $X^{monosomy}$ cells than in XX cells (Supplementary Fig. 2e–h).

To characterize the expression levels of genes subject to XCU in cells that do not undergo XCU, we compared in XX cells the mean expression of genes defined as upregulated and non-upregulated. We found that genes that are upregulated in $X^{monosomy}$ are expressed at significantly higher levels in XX iPSCs compared to genes that are not subject to upregulation in $X^{monosomy}$ (Fig. 1f). This is in line with previous findings that highly expressed genes are also more likely to be dosage-sensitive[50], and that dosage compensation mechanisms have evolved primarily to protect dosage-sensitive genes from the harmful effects of gene dosage imbalances[51].

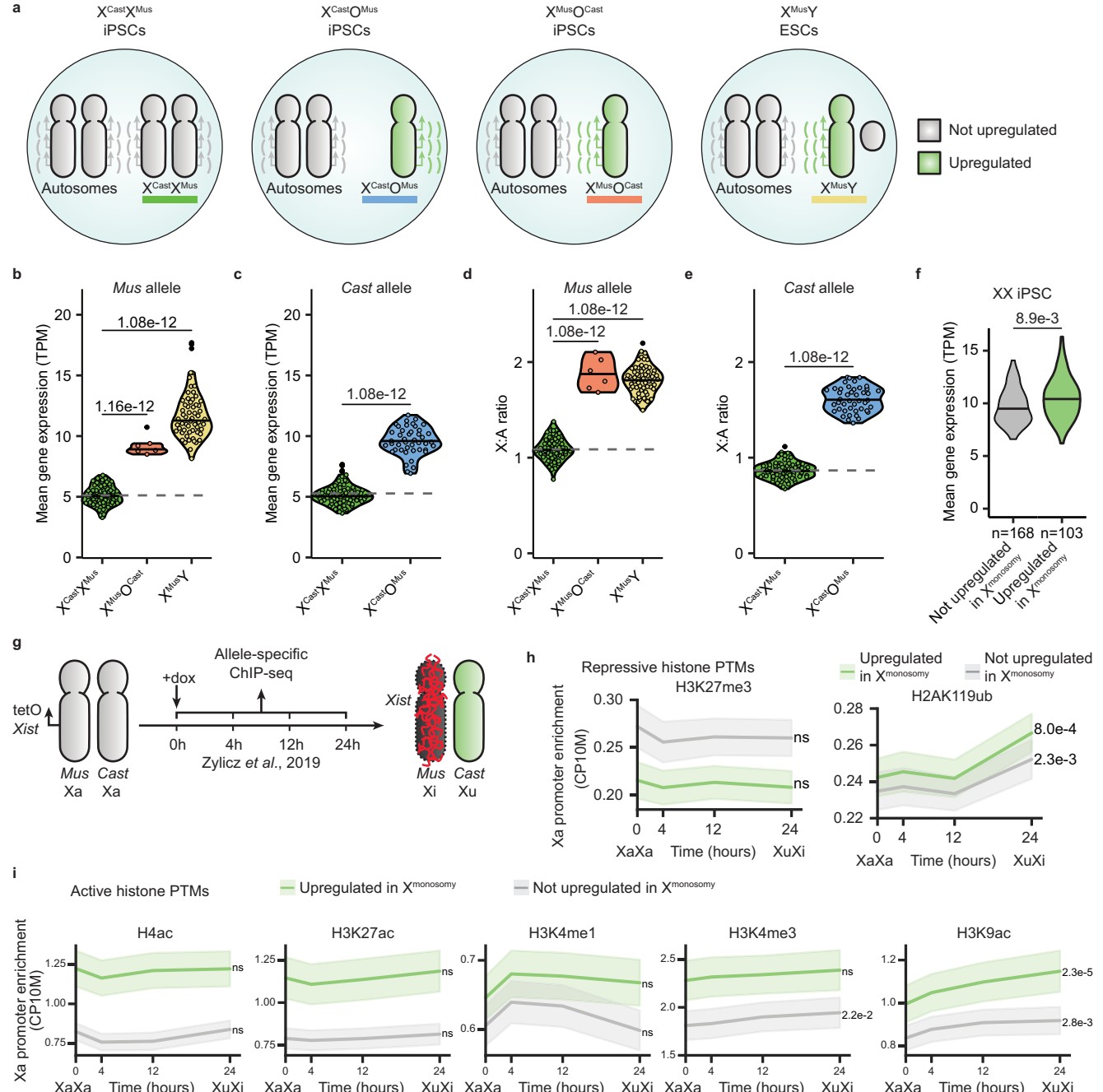

**Fig. 1 | Transcriptional dosage compensation of X-linked genes in mouse pluripotent stem cell lines with X-chromosome monosomy. a** Schematic depiction of cells with different sex chromosome complements (XX, XY, and XO) and the corresponding expected upregulation status of the X chromosome (gray: not upregulated, green: upregulated). **b, c** Violin plots of the mean normalized gene expression from the *Mus* (**b**) and *Cast* (**c**) allele of all expressed X-linked genes in X^Cast^X^Mus^ (green), X^Cast^O^Mus^ (blue), X^Mus^O^Cast^ (orange), and X^Mus^Y (yellow) cells. Dashed line indicates the mean value of XX cells, serving as a reference line for expected expression without upregulation. Each dot represents a cell. *P*-values were calculated via the Tukey HSD test (two-sided, with Tukey–Kramer correction for multiple comparisons). **d, e** Violin plots of the mean X:A ratio of the *Mus* (**d**) and *Cast* (**e**) allele in X^Cast^X^Mus^, X^Cast^O^Mus^, X^Mus^O^Cast^, and X^Mus^Y cells. The dashed line indicates the mean value of XX cells, serving as a reference line for expected expression without upregulation. Each dot represents a cell. *P*-values were calculated using the Tukey

HSD test (two-sided, with Tukey–Kramer correction for multiple comparisons). **f** Normalized expression in X^Cast^X^Mus^ iPSCs of genes annotated as upregulated (in XO and XY cells, *n* = 103, green) or non-upregulated (*n* = 168, gray). *P*-value was calculated using a two-sided Student's *t*-test. **g** Schematic overview of the experimental set-up used to generate allele-specific ChIP-seq data[57]. **h, i** Line plots showing the promoter enrichment of repressive (**h**) and active (**i**) histone modifications on the active *Cast* allele during XCI inactivation, and thus XCU induction. Promoter enrichment was quantified as the mean counts per 10 million (CP10M) in the 1 kb region upstream of the TSS (as shown for the 24 h time point in Supplementary Figs. 3d and 4c). Mean enrichment is plotted for the promoters of upregulated (green) and non-upregulated (gray) genes separately, with shaded areas indicating the standard error. *P*-values comparing enrichment at 0 and 24 h were calculated using two-sided paired *t*-tests, corrected for multiple testing with the Benjamini–Hochberg method, and shown to the right of each gene set.

Interestingly, gene ontology analysis indicated that upregulated X-linked genes were enriched for critical basic cellular functions such as regulation of chromatin organization and positive regulation of nucleic-acid templated transcription, whereas non-upregulated genes were enriched for specific developmental processes such as visual perception and negative regulation of axonogenesis, processes that are irrelevant at this stage of development (Supplementary Fig. 2i). Proteins encoded by upregulated genes were more likely to have protein-protein interactions (PPIs) with transcription factors, which could be due to feedback loops which allow upregulated genes to self-regulate their expression (Supplementary Fig. 2j)[52]. Genes encoding proteins that form complexes have been hypothesized to be particularly dosage-sensitive, potentially requiring gene dosage compensation in order to maintain the correct stoichiometry for proper complex function[51,53]. However, despite an apparent trend, our upregulated X-linked genes did not contain a significantly higher proportion of genes which code for proteins in complexes when compared to non-upregulated X-linked genes (Supplementary Fig. 2k). Examples of important complexes containing subunits encoded by upregulated genes include the MLL1, MSL, NSL, and DNA polymerase complexes, which contain subunits respectively encoded by *Hcfc1*, *Msl3*, *Ogt*, and *Pola1* genes.

Proteins which form condensates have been proposed to be particularly dosage-sensitive[54]. However, we found no significant difference in the proportion of proteins forming condensates between upregulated and non-upregulated genes, indicating that transcriptionally upregulated X-linked genes may not be enriched for genes encoding proteins known to form condensates (Supplementary Fig. 2l). Given the challenges in studying condensates, it remains possible that not all condensate-forming proteins have been identified, and future discoveries may revise this conclusion[55]. Furthermore, we hypothesized that genes which escape XCI, and thus retain expression from both alleles, require dosage compensation in X^monosomy cells. We observed a non-significant trend towards an increased proportion of upregulated genes that escape XCI (Supplementary Fig. 2m), potentially due to the limited statistical power from the small escapee gene set (n = 38)[56]. Nearly all orthologs of the upregulated X-linked genes were associated with diseases in the OMIM database; however, this proportion was not significantly higher than that of the non-upregulated genes (Supplementary Fig. 2n). Taken together, these results suggest that upregulated genes may be dosage-sensitive, and this may be linked to high levels of gene expression, involvement in protein complexes, and important cellular function.

## Histone post-translational modifications and XCU

To explore a potential link between histone post-translational modifications (hPTMs) and XCU, we analyzed hPTMs in a system that enables conditional induction of XCU in mouse XX ESCs[57]. In the study of Zylicz et al.[57], a doxycycline inducible promoter was inserted upstream of the endogenous *Xist* gene on the *Mus* allele in polymorphic (X^Cast X^Mus) mESCs[57]. Addition of doxycycline for 24 h activates *Xist* on the *Mus* allele, inducing XCI *in cis*, and triggers XCU on the opposite active X^Cast chromosome in trans[20]. Therefore, this experimental system enables us to examine hPTMs during XCU. To assess the enrichment of various hPTMs, we examined native ChIP-seq data with allele-resolution for several hPTMs (H4ac, H3K27ac, H3K4me1, H3K4me3, H3K9ac, H3K27me3, H2AK119ub) at 0, 4, 12, and 24 h of XCI and XCU induction with allele-resolution (Fig. 1g). As expected, during XCI, there was an increase in repressive marks H3K27me3 and H2AK119ub at the promoter region and across the gene body of X-linked genes on the Xi (Supplementary Fig. 3a–c). By contrast, on the Xa during XCU, there was no increase in the repressive mark H3K27me3 and minor increase in H2AK119ub in the promoter regions (Fig. 1h and Supplementary Fig. 3d). Across the gene body, there was a significant decrease in H3K27me3 and a small but protracted increase in H2AK119ub on the Xa during XCU (Supplementary Fig. 3d-e).

Additionally, active marks decreased in the promoter region of genes on the Xi (Supplementary Fig. 4a, b). However, on the Xa, H4ac, H3K27ac, H3K4me1, and H3K4me3 remained high, while H3K9ac significantly increased during XCU (Fig. 1i and Supplementary Fig. 4c). These results indicate that while active marks become depleted during XCI, a subset of active hPTMs become more enriched during XCU, or remain high (Supplementary Fig. 4d). Our results implicate these hPTMs in XCU in mammals in contrast to the recent view that they are not enriched on the Xu[20,25,29,41].

In addition, we found that before XCU, all active marks studied are more highly enriched on genes that will become upregulated during XCU versus those that do not, in line with the higher expression levels of these genes before XCU (Fig. 1i and Supplementary Fig. 4c). Unlike a previous study[20], we find evidence for an increase in several active hPTMs during XCU. This could be because the changes we detected are restricted to the promoter region. A previous study did propose increased hPTM on the Xu in XX mESCs[25], but given we now know XX mESCs have no XCU, the conclusions from this study must be revisited. By contrast, strong evidence has been provided for a role of H4K16ac in XCU in *Drosophila melanogaster*, where the male-specific lethal (MSL) complex deposits H4K16ac on the Xu[58]. To investigate if mammals and *Drosophila* both acquired mechanisms to enrich H4K16ac on the Xu, we investigated whether H4K16ac is enriched on the Xu in mice. A previous study analyzed acetylated H4 (H4ac) in mESCs but did not identify any enrichment on the Xu, the pan-H4ac detection potentially masking H4K16ac enrichment[57]. We therefore performed H4K16ac ChIP-seq in both XX and XO mESCs, which showed no significant increase in H4K16ac on the Xu compared to the Xa (Supplementary Fig. 4e, f). In summary, we found evidence for enrichment of active hPTMs during XCU in mammals similar to *Drosophila*. However, different sets of histone marks appear to be enriched on the Xu in different species, H4K16ac in *Drosophila*[58], and H3K9ac in mice.

## No evidence for an XCU center

To gain insights into how cells sense the number of active X chromosomes they carry and subsequently induce XCU, we hypothesized the existence of a dedicated regulatory locus on the X chromosome, which we term the XCU center (Xuc). By analogy to the well-characterized X-inactivation center (Xic)[8,14,59], the Xuc is defined as an X-linked locus whose loss or inactivation on one allele leads to chromosome-wide upregulation of gene expression in trans on the other X chromosome (Fig. 2a). We propose that the presence of two active copies of the Xuc prevents XCU, whereas having only one active copy is necessary to induce XCU. Specifically, when both Xuc copies are active (two Xa), no XCU occurs; when one Xuc is active and the other is inactive or absent (one Xa), XCU is induced on the Xa. To test the Xuc hypothesis, we investigated the impact of deleting large segments on one of the two X chromosomes in XX mESCs on gene expression. We wondered if deleting a large segment or region on one of the two X chromosomes would be sufficient to induce chromosome-wide XCU in trans.

We used XX mESCs with large heterozygous segmental deletions on one of the two X chromosomes. The cell lines were (1) Deletion 1: a 49 Mb deletion between *Tfe3* and *Zic3*, (2) Deletion 2: a 15 Mb deletion between *Zic3* and *Dusp9*, and (3) Deletion 3: a 15 Mb deletion of the region between *Dusp9* and *NrOb1* (Fig. 2b and Supplementary Table 2)[41]. We also included X^Cast X^Mus iPSCs, X^Cast O^Mus iPSCs, X^Mus O^Cast iPSCs, and X^Mus Y ESCs as controls and performed Smart-seq3xpress scRNA-seq as above. After quality filtering, we obtained allele-resolution expression values for 405 X-linked and 12,016 autosomal genes in 325 cells. ScRNA-seq data confirmed the genotype of the cells (Fig. 2c). All cell lines expressed pluripotency markers, indicating that cells maintained a pluripotent stem cell identity (Supplementary Fig. 5a). Correlation analysis confirmed that the heterozygous deletions did not affect the transcriptome, as indicated by the strong correlation between all cell lines (Supplementary Fig. 5b). In summary,

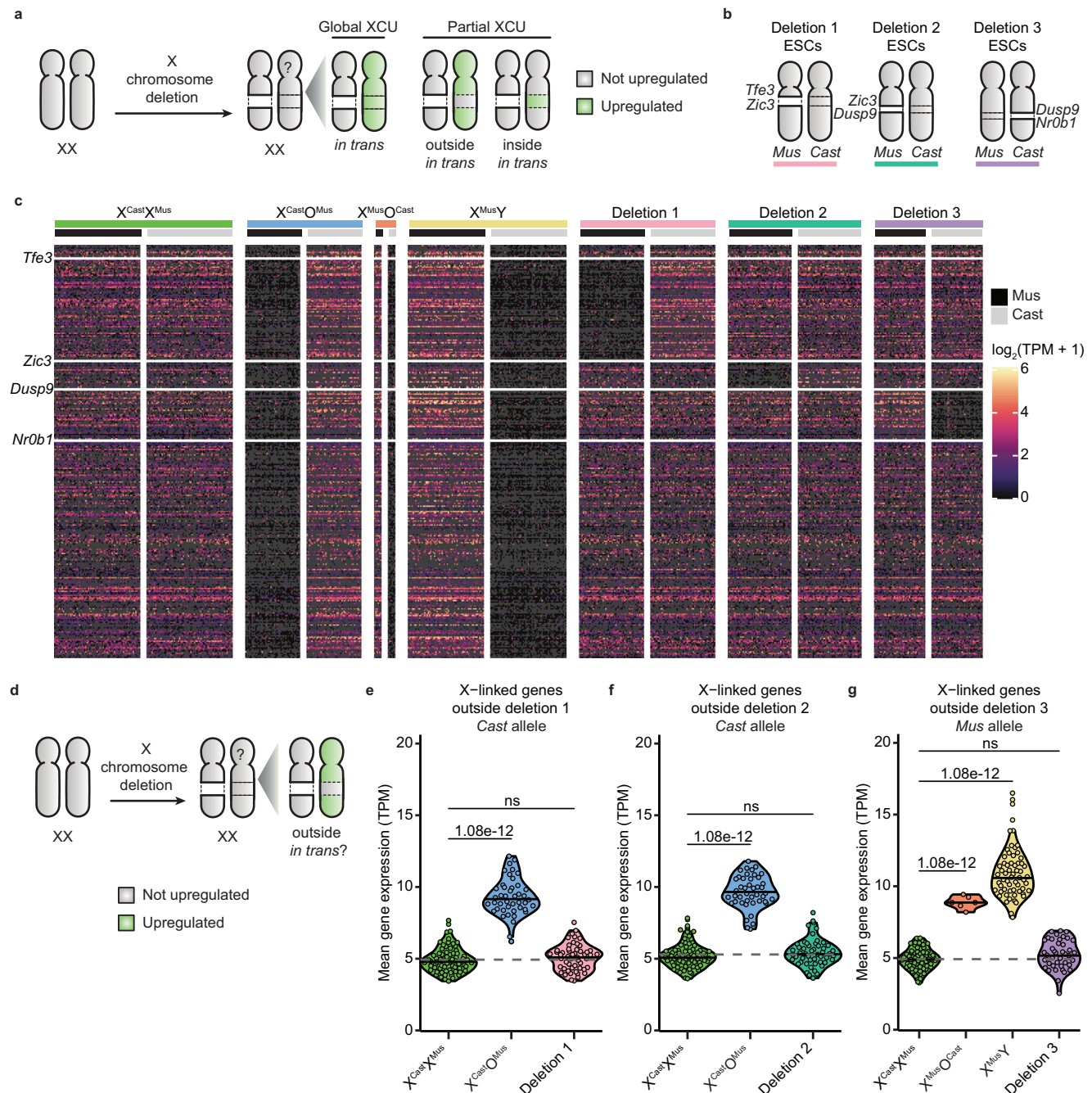

**Fig. 2 | Large heterozygous segmental deletions do not result in global XCU in trans. a** Schematic depiction of the XCU center hypothesis. Gray denotes an X chromosome that is not upregulated, while green indicates an upregulated X chromosome. **b** Schematic overview of the deletion cell lines used in the scRNA-seq experiment and their respective sex chromosome karyotypes. **c** Heatmap of all scRNA-seq data for X-linked genes, ordered by chromosomal position, grouped by cell line, and colored by log-transformed normalized expression values for each allele. The position of genes *Tfe3*, *Zic3*, *Dusp9*, and *Nr0b1* are highlighted, as these genes denote the start and end positions of each deletion. **d** Schematic depiction of

the XCU center hypothesis, such that deletion of a region of one X chromosome induces transcriptional upregulation for genes located outside of the deletion region in trans. **e–g** Violin plots separated by cell line showing the mean gene expression from the non-deleted allele for the genes located outside of the regions of Deletion 1 (**e**, *Cast* allele), Deletion 2 (**f**, *Cast* allele), and Deletion 3 (**g**, *Mus* allele). The dashed line indicates the mean value of XX cells, serving as a reference line for expected expression without upregulation. Each dot represents a cell. *P*-values calculated via the Tukey HSD test (two-sided, with Tukey–Kramer correction for multiple comparisons, ns not significant).

we generated allele-resolution scRNA-seq data for polymorphic PSCs with distinct heterozygous deletions of the X chromosome.

We first asked if deleting a large segment on one of the two active X chromosomes results in induction of global XCU in trans. Allele-specific gene expression analysis showed that none of the three deletions examined induced full global XCU in trans (Supplementary Fig. 5c–f). However, we noted a slight increase in gene expression in

trans in two cell lines, Deletion 1 and 3, but the magnitude of the increase was low. To understand where this increase may come from, we first analyzed gene expression specifically outside of the deleted segments in trans (Fig. 2d). While full loss of one of the two X chromosomes induces XCU, none of the segmental deletions induced full XCU in trans for genes located outside of the deletion segments (Fig. 2e–g and Supplementary Fig. 5g–j).

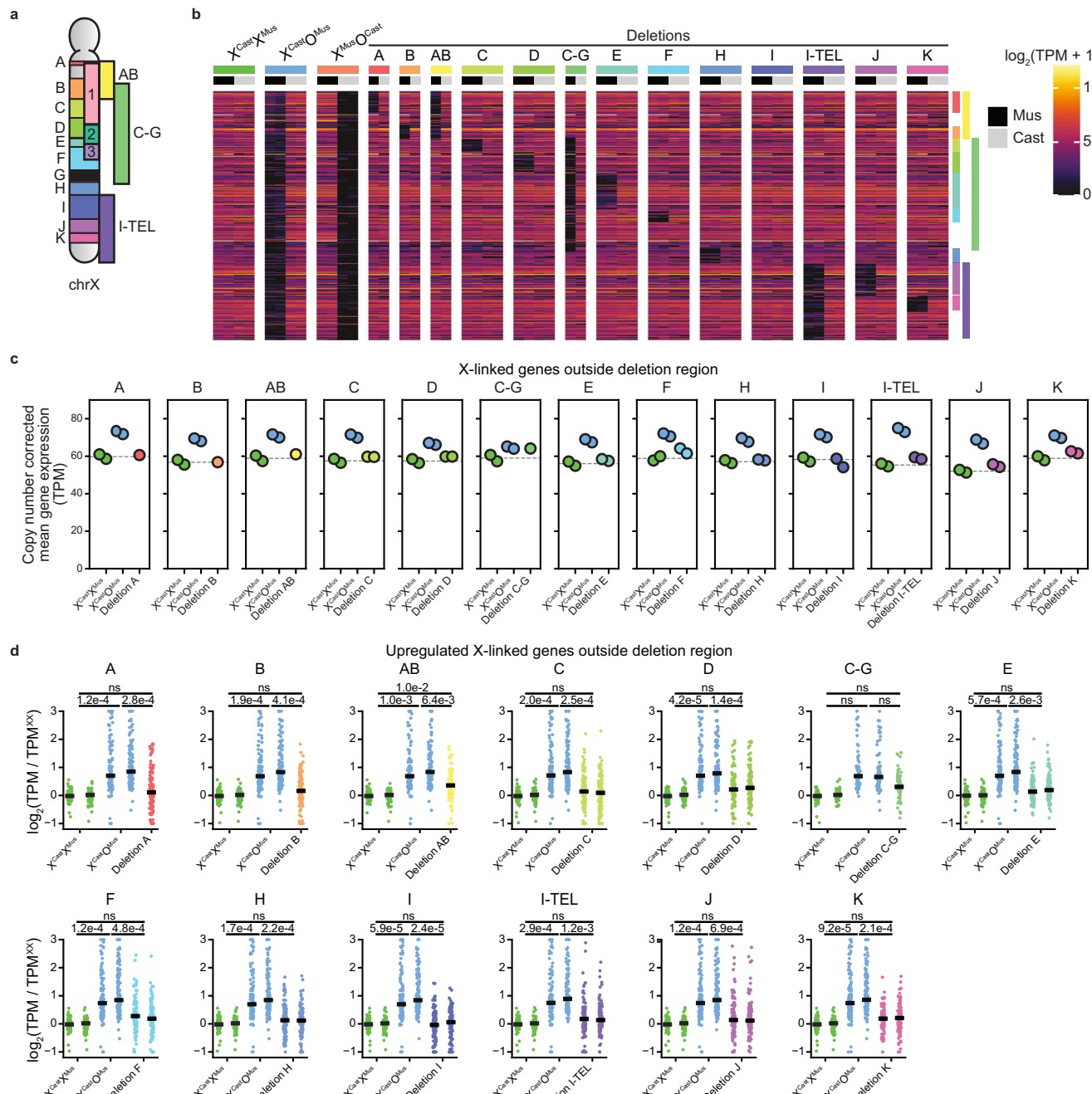

**Fig. 3 | The X chromosome does not contain an XCU center. a** Schematic overview of the deletions generated for the bulk RNA-seq experiment, as well as deletions 1, 2, and 3. All deletions were introduced in hybrid (*Mus/Cast*) ESC lines with a heterozygous *Rnf12* deletion on the *Mus* allele. Deletion G was found to be a recombination event rather than a deletion, and has been omitted from all subsequent analyses. **b** Heatmap of all X-linked genes ordered by chromosomal position, grouped by cell line, and colored by log-transformed normalized expression values for each allele. The expected deleted region for each deletion line is highlighted on the right side of the heatmap. **c** Scatter plots separated by deletion showing the mean copy number-corrected expression levels of genes located outside each deletion for $X^{Cast}X^{Mus}$, $X^{Cast}O^{Mus}$, and cells with the corresponding deletion. Each dot represents a biological replicate (independent clones). The dashed line represents the average level from the XX samples, serving as a reference line for expected expression without upregulation. Note that the gene sets differ between panels, resulting in varying expression levels in $X^{Cast}X^{Mus}$ and $X^{Cast}O^{Mus}$ across panels. **d** Scatter plots separated by deletion showing the log2 fold change in copy number-corrected expression levels of replicates of $X^{Cast}X^{Mus}$, $X^{Cast}O^{Mus}$, and cells with the corresponding deletion, relative to the mean $X^{Cast}X^{Mus}$ levels per gene. Only genes upregulated in the $X^{Cast}O^{Mus}$ cells and located outside each respective deletion are shown ($n$ = 110, 115, 93, 117, 112, 73, 113, 119, 118, 122, 95, 116, 116). Each dot represents a gene, and the solid line indicates the median log2 fold change. *P*-values comparing mean expression between genotypes were calculated using two-sided paired *t*-tests, and corrected for multiple testing with the Bonferroni method (ns not significant).

To determine if the conclusions can be extended to other regions of the X chromosome, we proceeded to test deletions that collectively span the entire X chromosome. We generated 13 new individual large segmental heterozygous deletions of the X chromosome in $X^{Cast}X^{Mus}$

mESCs (Fig. 3a, Supplementary Fig. 6a, and Supplementary Table 2). This included ten deletions, labeled A to K, as well as three larger deletions spanning multiple regions (AB, C-G, I-TEL). Deleted segments were chosen to respect syntenic segments and TAD boundaries[60] as

much as possible (Supplementary Fig. 6b). These deletions were generated in a $Rnf12^{+/-}$ parental cell line, which exhibits highly similar expression to wild-type XX mPSCs as expected (Supplementary Fig. 6c), and thus served as the reference sample for these analyses. We generated bulk RNA-seq data from two independent clones from most deleted lines. We used $X^{Cast}X^{Mus}$, $X^{Mus}O^{Cast}$, and $X^{Cast}O^{Mus}$ mESCs as controls. After quality filtering, we obtained expression values for 447 X-linked and 11,203 autosomal genes, enabling us to investigate X-linked gene dosage compensation. Allele-resolution analysis of X-linked gene expression confirmed the expected genotype of the cells (Fig. 3b). Pluripotency gene expression was maintained (Supplementary Fig. 6d). The heterozygous deletions did not affect the global transcriptome, as confirmed by a correlation analysis between all sequenced samples (Supplementary Fig. 6e). These results confirm the successful generation of a high-quality dataset of deletions spanning the entire X chromosome, suitable for testing the Xuc hypothesis.

As expected, XX cells had higher X-linked gene expression levels compared to XO cells (Supplementary Fig. 6f). However, when gene expression is normalized per X chromosome, XO cells showed XCU (Fig. 3c). In contrast, none of the 13 new deletions induced global full XCU for genes located outside of the deleted regions. At the gene level, genes upregulated in $X^{Cast}O^{Mus}$ cells remained expressed at lower levels outside the deletions across all deletion lines compared to $X^{Cast}O^{Mus}$ cells (Fig. 3d). None of the deletions caused significant expression changes relative to $X^{Cast}X^{Mus}$, with the exception of deletion AB. This likely represents an outlier, as deletion of region A or B alone had no significant effect, and the magnitude of change for AB was smaller than in $X^{Cast}O^{Mus}$ cells. In summary, deletion of large segments on one of the two X chromosomes is not sufficient to induce full XCU outside of the deleted regions. Collectively, these results do not support the hypothesis that any of the segments deleted act as an Xuc with the ability to induce global XCU of the X chromosome in trans.

## Cells sense large segmental deletions and compensate gene expression in trans

To test if large segmental deletions are compensated within the deleted regions in trans, we examined XCU specifically within deleted segments (Fig. 4a). Remarkably, there was a strong induction of XCU specifically within deleted segments in trans (Fig. 4b–d and Supplementary Fig. 7a–c). Moreover, the level of XCU was similar to that of XO cells in most cell lines. We made similar observations for the 13 segmental deletions of the X chromosome, in which XCU was induced within deleted segments in trans (Fig. 4e and Supplementary Fig. 7d). To quantify XCU at the gene level, we examined expression levels of genes upregulated in $X^{Cast}O^{Mus}$ cells relative to $X^{Cast}X^{Mus}$. These genes were consistently upregulated across the deletion lines, reaching expression levels similar to those in $X^{Cast}O^{Mus}$ cells (Fig. 4f). Allele-specific analysis confirmed upregulation of the Cast allele in $X^{Cast}O^{Mus}$ cells (Supplementary Fig. 7e). These results demonstrate that mammalian cells can sense when specific segments of one of the two X chromosomes are genetically deleted and compensate for gene dosage imbalances by transcriptional upregulation specifically in trans.

Next, we aimed to define if the upregulated genes within the deleted regions in the deleted lines are the same as the upregulated genes in XO cells. To address this, we asked if genes upregulated in XO cells were also upregulated when deleted in cells with large heterozygous deletions. We found that 64 of the 65 upregulated genes located within deletions 1–3 were also upregulated in each of the deletions in trans (Fig. 4g). Likewise, genes upregulated in cells with deletions were also upregulated in cells with $X^{monosomy}$ (Supplementary Fig. 7e). This high degree of overlap suggests a shared transcriptional response to X chromosome dosage loss.

These results favor a model in which specific regions or even individual genes can sense gene dosage and even gene expression states and adapt X-linked gene expression accordingly by XCU. We

investigated whether deletion of a single allele of a gene would also result in transcriptional dosage compensation of that gene in trans. In other words, we asked if one allele of a gene can sense and respond to deletion of the other allele in trans. We focused on a single gene, *Pdzd4*, that is subject to XCU, and generated mESCs which contain a heterozygous deletion of the promoter and gene body of *Pdzd4* on the *Mus* allele (Supplementary Fig. 7g). Allele-specific quantitative PCR analysis indicated upregulation in trans for three of the four clones heterozygous for *Pdzd4* (Fig. 4h). These results suggest that *Pdzd4* is transcriptionally compensated in trans when one of the two alleles of *Pdzd4* is deleted. We conclude that heterozygous loss of a gene body and corresponding promoter on the X chromosome can be sensed and compensated via increased transcription in trans.

## Autosomal gene dosage compensation by transcriptional upregulation in trans

So far, we have provided evidence for transcriptional compensation of monosomies of the X chromosome in trans, consistent with the evolution of dosage compensation mechanisms on the X chromosome by XCU[4–11]. Next, we aimed to determine if dosage compensation by transcriptional upregulation in trans can take place on autosomes. To explore autosomal gene dosage compensation, we used a bulk RNA-seq dataset of *Mus/Cast* mESCs with inducible autosomal gene silencing *in cis*[61]. In this system, addition of dox activates an inducible *Xist* transgene inserted on the *Cast* allele of chromosome 3, resulting in the transcriptional downregulation of many genes *in cis* on that allele (Fig. 5a and Supplementary Fig. 8a)[61]. This experimental setting enabled us to ask if reduction of autosomal gene expression *in cis* induces dosage compensation in trans. We confirmed reduced gene expression *in cis* and identified a ~ 80 Mb region with a high density of genes that decrease expression on the *Cast* allele after *Xist* expression for 24 h (Fig. 5b, c). To assess gene-by-gene compensation, we first examined genes located outside of the 80–160 Mb region in trans, on the *Mus* allele, and did not detect changes in gene expression (Fig. 5d). By contrast, inducing reduced gene expression of chromosome 3 *in cis* induced significant transcriptional upregulation of the corresponding *Mus* allele in trans in the 80–160 Mb region (Fig. 5e). Contrary to our findings on XCU, the baseline expression level prior to addition of doxycycline was significantly lower for upregulated genes compared to non-upregulated genes (Supplementary Fig. 8b), potentially due to the fact that the expression of highly expressed genes is not sufficiently reduced in this experimental system. Upregulated chromosome 3 genes were not enriched for genes which encode proteins in complexes, or proteins which form condensates (Supplementary Fig. 8c, d). In summary, we found that decreasing expression *in cis* resulted in upregulation in trans for several genes on chromosome 3 after 24 h of doxycycline. Therefore, these results suggest that reducing expression of autosomal genes *in cis* can be compensated by transcriptional upregulation in trans.

## Gene dosage compensation at the protein level

XCU has been described to occur at the transcriptional level in mouse embryos and ESCs, but its occurrence at the protein level in mouse PSCs remains to be established[16,17,20,62]. To determine if XCU takes place at the protein level in XO and XY ESCs, we performed quantitative proteomics in $X^{Cast}X^{Mus}$, $X^{Mus}Y$, and $X^{Cast}O^{Mus}$ ESCs. We also included two ESC lines with large heterozygous segmental deletions (Deletions E and K), which showed high percentages of transcriptionally upregulated genes located within the deleted region (Supplementary Table 2). We identified 8050 proteins, of which 4446 proteins were detected in all samples.

We next analyzed X-linked protein levels and found them to be more abundant in $X^{Cast}X^{Mus}$ ESCs compared to $X^{Mus}Y$ and $X^{Cast}O^{Mus}$ cells, in agreement with increased X-linked gene expression in these cells due to the presence of two active X chromosomes (Supplementary

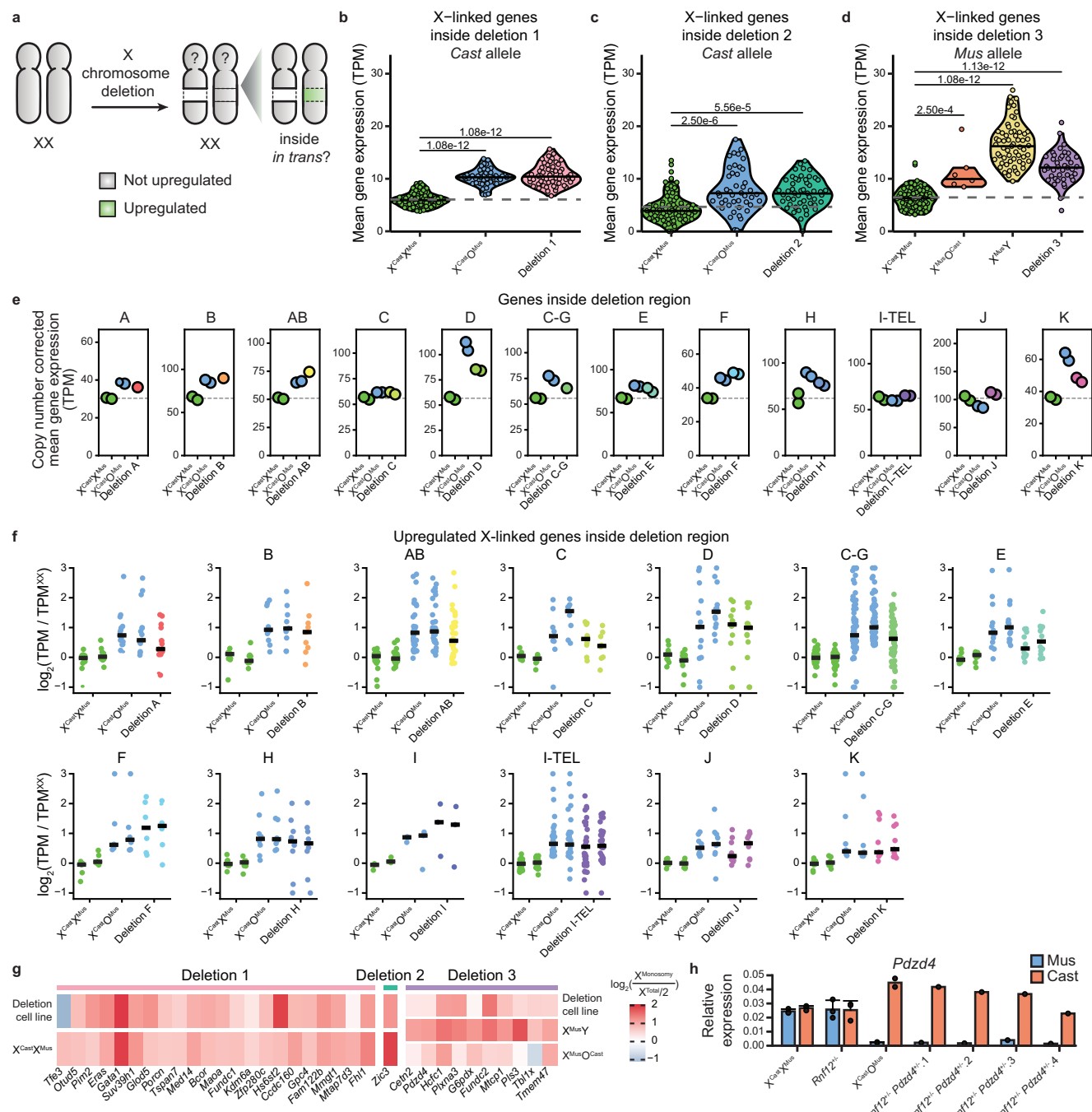

**Fig. 4 | Dosage compensation by transcriptional upregulation occurs in trans for genes within regions of X-chromosome monosomy. a** Schematic depiction of gene-by-gene regulation of dosage compensation. Gray denotes an X chromosome that is not upregulated, while green indicates an upregulated X chromosome. **b**–**d** Violin plots separated by cell line showing the mean gene expression from the non-deleted allele for the genes located inside the regions of Deletion 1 (**b**, *Cast* allele), Deletion 2 (**c**, *Cast* allele), and Deletion 3 (**d**, *Mus* allele). The dashed line indicates the mean value of XX cells, serving as a reference line for expected expression without upregulation. Each dot represents a cell. *P*-values were calculated via the Tukey HSD test (two-sided, with Tukey–Kramer correction for multiple comparisons). **e** Scatter plots separated by deletion showing the mean copy number-corrected expression levels of genes located within each deletion for $X^{Cast}X^{Mus}$, $X^{Cast}O^{Mus}$, and cells with the corresponding deletion. Each dot represents a biological replicate (independent clones). The dashed line indicates the average level from the XX samples, serving as a reference line for expected expression without upregulation. Deletion I did not contain sufficient expressed genes and has been omitted. **f** Scatter plots separated by deletion showing the log$_2$ fold change in copy number-corrected expression levels of replicates of $X^{Cast}X^{Mus}$, $X^{Cast}O^{Mus}$, and cells with the corresponding deletion, relative to the mean $X^{Cast}X^{Mus}$ levels per gene. Only genes upregulated in the $X^{Cast}O^{Mus}$ cells and located within each respective deletion are shown ($n = 15, 10, 32, 8, 13, 52, 12, 6, 7, 3, 30, 9, 9$). Each dot represents a gene, and the solid line indicates the median log$_2$ fold change. **g** Heatmap showing the fold change in expression of upregulated genes identified from the allelic single-cell analysis between $X^{monosomy}$ and $X^{Cast}X^{Mus}$. Fold change is calculated between *Cast* expression from either $X^{Cast}O^{Mus}$, Deletion 1, or Deletion 2 cells and total allelic expression divided by 2 from XX cells, or *Mus* expression from $X^{Mus}O^{Cast}$, $X^{Mus}Y$, or Deletion 3 cells and total allelic expression divided by 2 from $X^{Cast}X^{Mus}$ cells. **h** Bar plot of the relative RNA expression of the gene *Pzdz4* in XX wild type ($X^{Cast}X^{Mus}$, $n = 3$), *Rnf12*$^{+/-}$ parental line (*Rnf12*$^{+/-}$, $n = 3$), XO ($X^{Cast}O^{Mus}$, $n = 2$), and 4 different *Rnf12*$^{+/-}$*Pzdz4*$^{+/-}$ clones for both the *Mus* (blue) and *Cast* (red) allele. Average expression ± standard deviation, each dot represents a biological replicate (for XO and *Rnf12*$^{+/-}$*Pzdz4*$^{+/-}$ from independent clones).

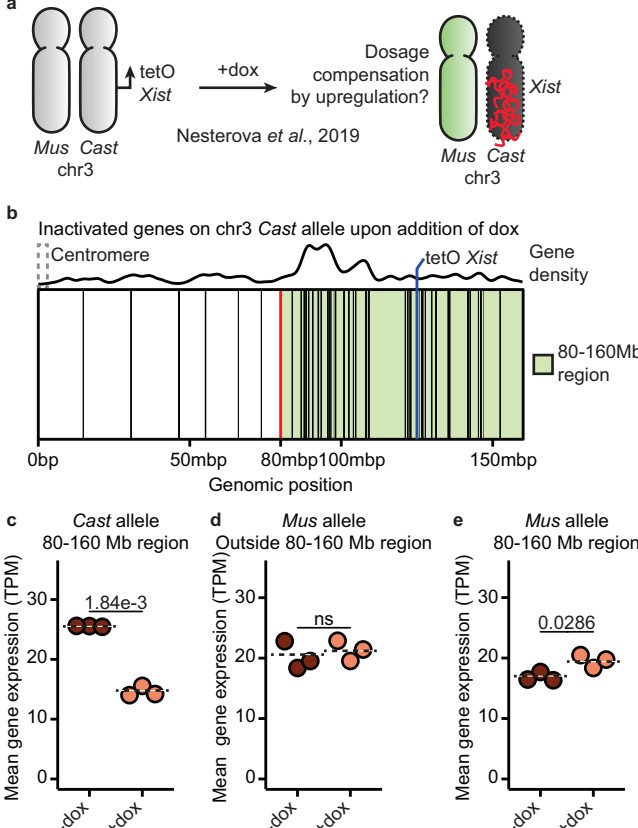

**Fig. 5 | Dosage compensation is induced by transcriptional upregulation in trans within regions of monosomic expression on chromosome 3. a** Schematic depiction of the location of the *Xist* transgene on chromosome 3[61]. After 24 h of dox treatment, *Xist* coats the *Cast* chromosome 3, reducing gene expression. **b** Overview of the genomic location of genes showing reduced expression on the Cast allele of chromosome 3 (black lines), as well as the location of the left border (red line) of the region defined as being inactive (green area) and the site of the *Xist* transgene (blue line). **c–e** Plot of median normalized expression from the *Cast* (**c**) or *Mus* (**d**, **e**) allele of genes inside (**c**, **e**) or outside (**d**) the region with reduced gene expression for both control (−dox) and treated (+dox) conditions. Dashed lines indicate the mean levels per condition. *P*-values were calculated via the two-sided Student's *t*-test (ns not significant). Each dot represents a biological replicate.

Fig. 9a). Since mass spectrometry results do not provide allelic resolution, we normalized protein abundances for the X-chromosome copy number. On a per-X-chromosome basis, XCU was detected at the protein level in X$^{Mus}$Y and X$^{Cast}$O$^{Mus}$ ESCs compared with X$^{Cast}$X$^{Mus}$ ESCs (Fig. 6a). Indeed, ESCs with one active X chromosome showed more than half of X-linked protein abundances, suggesting XCU at the protein level (Supplementary Fig. 9a). The increase was also seen for the X:A ratio (Supplementary Fig. 9b). Therefore, we confirm that XCU takes place at the protein level in X$^{Mus}$Y and X$^{Cast}$O$^{Mus}$ ESCs to compensate for the absence of one of the two active X chromosomes. These results show that the single active X chromosome of mESCs is upregulated at the protein level compared to each X chromosome in XX mESCs.

We next examined which proteins are subject to upregulation in ESCs with only one active X chromosome. Focusing on the 159 detected X-linked proteins, 45 proteins (38%) were significantly upregulated in both X$^{Mus}$Y and X$^{Cast}$O$^{Mus}$ cells, indicating that some but not all expressed proteins are compensated by XCU (Fig. 6b, c, Supplementary Fig. 9c, d, and Supplementary Table 3). Compared to autosomes, the proportion of upregulated proteins was highly enriched on the X chromosome (Fig. 6d). At the protein level, upregulated X-linked

proteins were enriched for proteins that are part of complexes, suggesting evolution selected dosage compensation mechanisms for these proteins (Fig. 6e). Taken together, our data suggest that a subset of X-linked genes are compensated by XCU at the protein level in X$^{Mus}$Y and X$^{Cast}$O$^{Mus}$ mESCs.

To determine if the same genes are compensated at the RNA and protein level, we integrated the proteomics and transcriptomics data and compared genes only present in both RNA and protein data. We found that of the 159 coding genes covered in both datasets, a majority, 103 (60%), were more abundant in XO samples compared to the copy-number-corrected abundance in XX (Fig. 6f). Specifically, 47 (28%) showed both increased RNA and protein levels, whereas 56 (35%) genes were increased at the protein level only, suggesting exclusive post-transcriptional dosage compensation of these genes. A small proportion of genes were increased at the RNA level only (8%), while 27% of genes did not show compensation at the RNA or protein level. Despite the limited sensitivity of proteomics compared to transcriptomics and the commonly reported RNA-protein discordance[63], we observed a positive correlation in upregulation ($r = 0.37$). These results indicate that the contribution of RNA and protein regulation to dosage compensation by XCU may vary for different X-linked genes in mESCs, in agreement with transcription and protein measurements in the brain[62]. Our results reveal that there are multiple layers of gene and protein regulation involved in dosage compensation by XCU of individual X-linked genes in ESCs.

Next, we examined dosage compensation at the protein level in two heterozygous segmental deletion lines E and K. To assess XCU, we analyzed protein abundance normalized to copy number, based on information on genetic regions with one or two copies. Outside the deleted regions, there was no XCU at the protein level (Fig. 6g and Supplementary Fig. 9e). In contrast, within the deleted regions, we detected considerable XCU at the protein level (Fig. 6h and Supplementary Fig. 9f). The upregulation status of individual proteins that are upregulated in X$^{Mus}$Y and X$^{Cast}$O$^{Mus}$ lines was evaluated in the deleted ESC lines (Fig. 6i). Proteins encoded outside the deleted regions remained unchanged, while several proteins within deleted regions were upregulated. In summary, our results reveal the existence of different dosage compensation mechanisms in mouse PSCs, with genes that tend to be compensated at the protein, RNA or both levels, and to varying degrees. Together, we conclude that ESCs can compensate for mono-allelic gene expression by XCU at both RNA and protein levels.

## Discussion

In this study, we investigated dosage compensation and sensing in PSCs. We discovered that dosage of monosomically expressed X-linked and autosomal regions is compensated at both the mRNA and protein levels. In addition, we found that dosage compensation does not require one or more specific regions on the X chromosome involved in dosage sensing and XCU. In contrast, our findings indicate that XCU is regulated at a gene-by-gene level and involves upregulation of expression at the transcriptional and protein levels.

These results support previous findings by others and us related to elastic tuning of dosage compensation with one allele becoming upregulated upon inactivation or loss of the other allele during development[16,20]. In addition, our studies are in line with reports showing that XCU is not global, as not all X-linked genes are transcriptionally upregulated[17]. Our data do indicate that a majority of proteins detected by mass spectrometry display XCU at the protein level. This finding may involve an over-interpretation of the percentage of genes subject to XCU as our findings indicate that XCU genes are expressed at a higher level and will therefore be more easily detected. On the contrary, our study only examined PSCs and different cell types or cell states may require dosage compensation of different

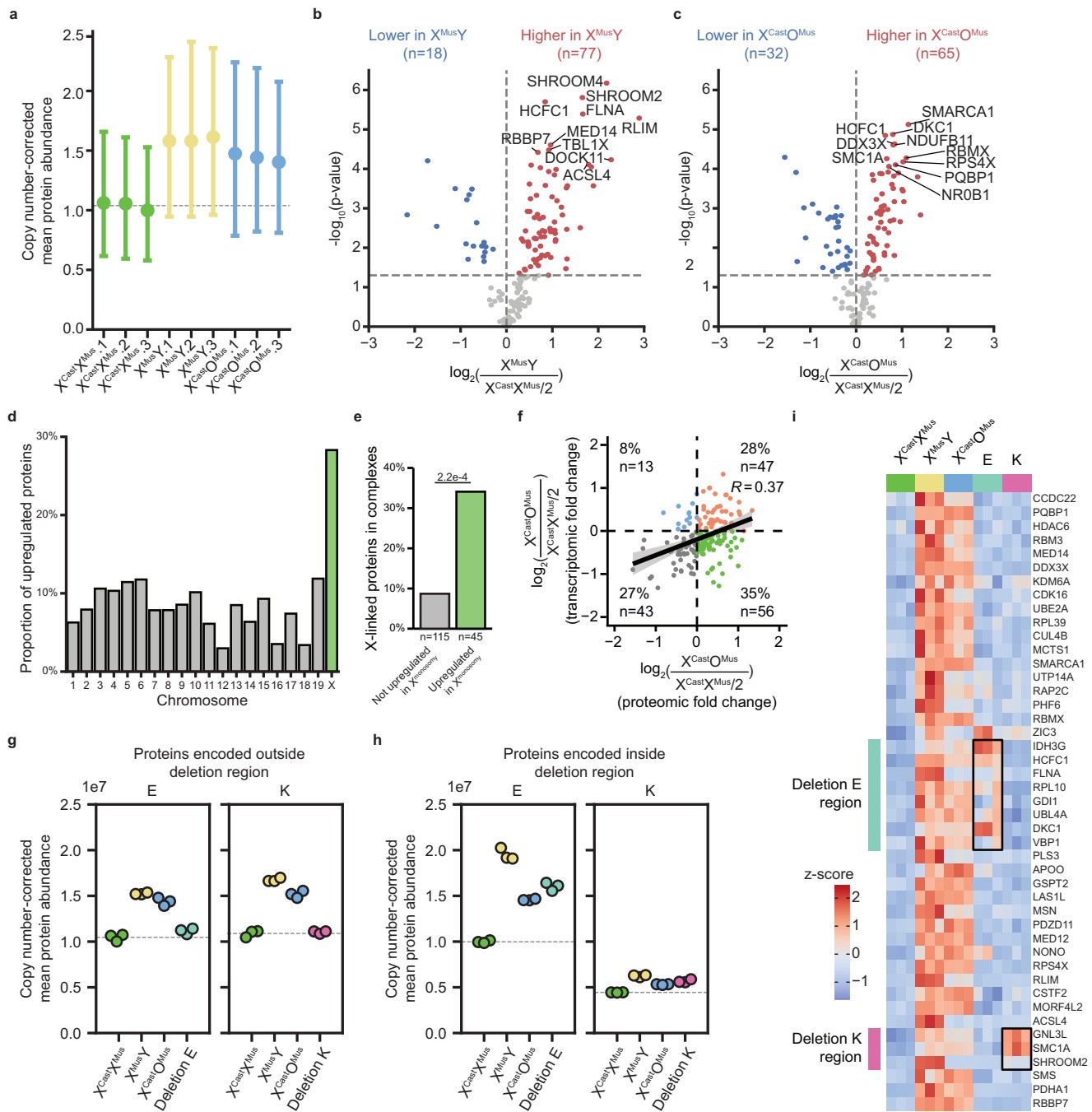

**Fig. 6 | Protein-level dosage compensation is induced within regions of monosomic expression on the X chromosome. a** Copy number-corrected protein abundance for all X-linked proteins in $X^{Cast}X^{Mus}$, $X^{Mus}Y$ and $X^{Cast}O^{Mus}$ ESCs. The mean (of $n = 127$ proteins) per biological replicate from a single clone is depicted with a dot, with the corresponding 95% confidence intervals shown as error bars. The dashed line indicates the mean value of $X^{Cast}X^{Mus}$ cells, serving as a reference line for expected expression without upregulation. **b, c** Volcano plots displaying the differential protein expression analysis of X-linked proteins between $X^{Mus}Y$ and $X^{Cast}X^{Mus}$ samples (**b**) and between $X^{Cast}O^{Mus}$ and XX samples (**c**) ($n = 3$). Each dot represents a protein. $P$-values were calculated from copy number-corrected protein quantifications using Tukey's HSD test (two-sided, with Tukey–Kramer correction for multiple comparisons). Significantly up and downregulated proteins are highlighted in red and blue, respectively ($\alpha < 0.05$, $\log_2$(fold change) > 0.5 or < −0.5, respectively). **d** Barplot displaying the proportion of upregulated genes per chromosome. The X chromosome is highlighted in green. **e** Barplot showing the proportion of complex-forming proteins in the upregulated (green) and non-

upregulated (gray) protein lists. $P$-values were calculated via the Chi-square test. **f** Scatter plot showing the correlation between upregulation at RNA and protein level. Each dot represents a gene/protein. For both levels, the fold change between $X^{Cast}O^{Mus}$ and half of the $X^{Cast}X^{Mus}$ level is shown for each gene. Genes are colored according to their upregulation status at both (orange), either (blue or green) or neither (gray) levels. Shaded area indicates 95% confidence interval. **g, h** Scatter plots separated by region showing the mean copy number-corrected abundance levels of proteins encoded by genes located outside (**g**) and within (**h**) region E and K for $X^{Cast}X^{Mus}$, $X^{Mus}Y$, $X^{Cast}O^{Mus}$ ESCs, and ESCs with the corresponding deletion. Each dot represents a biological replicate from a single clone. The dashed line represents the average abundance levels from the $X^{Cast}X^{Mus}$ samples. **i** Heatmap showing the $z$-scores of copy-number-corrected protein quantifications for X-linked proteins upregulated in both $X^{Cast}Y$ and $X^{Cast}O^{Mus}$ samples compared to $X^{Cast}X^{Mus}$. Proteins are ordered by genomic locations, with deletions E and K highlighted in light blue and pink, respectively. Proteins expected to be upregulated are highlighted with a box.

genes[17,62,64]. We conclude that the mammalian genome is more adaptive and robust than previously suggested.

Our studies and others also indicate that different dosage compensation mechanisms or different combinations of mechanisms are in place to regulate gene dosage compensation. These mechanisms include transcriptional burst and translation initiation frequency as well as mRNA and protein stability[19,26,62]. We hypothesize that self-regulating feedback loops and enhancers may play a role in sensing changes in gene dosage, as inactivation or deletion of one allele could drive differential transcription factor binding in trans. This could then also be related to the increased transcriptional burst frequency, which has been reported with XCU[16,19,20]. Furthermore, in line with the increased transcriptional burst frequency reported, we found an increased enrichment of one histone modification, H4K9ac, associated with active transcription at the transcription start sites of genes subject to XCU. These results show that H4K9ac is associated with XCU; however, whether this mark is instructive or sufficient to induce dosage compensation remains to be tested. Notably, H4K16ac was not significantly enriched, indicating that mammalian cells employ distinct mechanisms compared to *Drosophila* to upregulate expression of X-linked genes. As our study indicates that XCU is regulated on a gene-by-gene basis, identification of the mechanisms employed and interactions involved to regulate XCU requires further investigation at the gene level.

Studies in humans and primates have suggested that XCU also occurs in cells with a single active X chromosome[27,28,35,62]. We hypothesize that the gene-specific dosage sensing and subsequent compensation observed in mouse cells also occurs in human stem cells and in development. Furthermore, it has been suggested that XCU is widespread in placental mammals[4], so we would expect gene-specific dosage compensation to occur for many genes on the human X chromosome; however, additional studies would be required to confirm this. Widespread dosage compensation of X-linked genes could explain why monosomy of the X chromosome (Turner syndrome) is the only viable monosomy in humans and mice[65,66]. Studying X-linked dosage compensation in early human development is difficult without allele-resolution data, and due to the low coverage of SNPs between human alleles, allele-resolution data is typically only available for a few genes.

Our findings have implications for human disorders caused by haploinsufficiency. First, dosage compensation of autosomes may have evolved to cope with heterozygous mutations, monoallelic silencing, microdeletions, or aneuploidies and their own effects. Second, dosage compensation may enable survival, but is not sufficient for restoring a wildtype phenotype and rescuing haploinsufficiency and as such may enable diseased states.

Consistent with previous research in both human cancers and embryos, our findings provide strong evidence for dosage compensation of autosomal genes[3,67]. This suggests that the mammalian genome possesses an inherent ability to compensate for gene dosage imbalances resulting from genomic loss or decreased expression, a capacity that extends beyond the X chromosome. This intrinsic dosage compensation mechanism may have been a key factor in the evolution of XCI. As X-linked genes were progressively lost from the Y chromosome during the evolution of mammalian sex chromosomes[68], it is hypothesized that increasing selective pressure drove the development of dosage compensation mechanisms for X-linked genes. This likely led to the gradual incorporation of X-linked genes into the XCI process[8,34,69]. The gradual nature of this evolution may explain the gene-by-gene regulation observed in XCU.

In conclusion, our study supports the hypothesis of gene-by-gene regulation of X-linked gene dosage compensation involving multiple levels of regulation in a locus-specific manner to sense and compensate for variations in X-linked gene dosage.

## Methods

### Generation of cell lines

XX female polymorphic mESCs (129/Sv-Cast/EiJ) with heterozygous deletions located between *Tfe3* and *Zic3*, *Zic3* and *Dusp9*, as well as between *Dusp9* and *Nr0b1* were obtained from Song et al., 2019[41]. XO mouse iPSCs used in scRNA-seq were generated by passaging XX-GFP female mouse iPSCs (129/Sv-Cast/EiJ, obtained from Talon et al., 2021[16]) to passage 14 before FACS for GFP-negative cells (XO iPSCs) and collection for scRNA-seq (see below). XX female iPSCs (129/Sv-Cast/EiJ) were derived as described in Talon et al., 2021[16]. XY male mESCs (C57BL/6J-Cast/EiJ) were obtained from Chen et al., 2016[70]. Both ESC and iPSC lines were used in this study, due to availability and experimental design constraints; however, evidence indicates that mouse iPSCs and ESCs behave equivalently with respect to X chromosome sensing and XCU[16,20].

For all newly generated lines, gRNAs (Supplementary Table 4) were integrated in the pSpCas9(BB)−2A-Puro V2.0 PX459 vector (Addgene #62988), enabling transient puromycin selection. Complementary oligonucleotides containing the gRNA sequence and compatible *BbsI* overhangs were annealed and ligated in PX459. To generate homology arm vectors, -500–1000 bp 5′ and 3′ homology arms were amplified by overhang PCR from mouse genomic DNA with compatible restriction enzyme sequences. PCR products were purified from gel using the QIAquick® Gel Extraction Kit (Qiagen, 28506). Homology arm PCR products were digested with the indicated restriction enzymes and subsequently purified and ligated into compatible sites in the pCR™2.1-TOPO™ vector (Invitrogen, K450002). A neomycin resistance cassette flanked by LoxP sites was integrated in-between the 5′ and 3′ homology arms in an EcoRI restriction site for generating large X-chromosomal deletions. The cassette was retained in all deletion lines, as Cre-mediated excision was not required for downstream experiments.

Two independent *Rnf12*$^{+/-}$ cell lines were generated in F1 2−1 hybrid (129/Sv-Cast/Ei) mESCs. Although the heterozygous *Rnf12* mutation was not essential for the deletion experiments described here, it induces complete skewing upon XCI, which was relevant for a parallel study. The first *Rnf12*$^{+/-}$ cell line was generated by introducing PX459 vectors containing gRNAs targeting intron 2 and the 3′UTR. This *Rnf12*$^{+/-}$ cell line served as the parental cell line for generating deletions A through I and K, as well as the *Rnf12*$^{+/-}$*Pdzd4*$^{+/-}$ lines. Attempts to induce deletion J in this line failed due to chromosomal instability and subsequent X chromosome loss. Therefore, a second *Rnf12*$^{+/-}$ cell line was generated by introducing PX459 vectors with alternative gRNAs. This line served as the parental background for del J and I-TEL.

The first *Rnf12*$^{+/-}$ was generated by transfecting gRNAs by electroporation in Gene Pulser 0.2 cm-gap cuvettes (Biorad, 1652086) at 118 kV, 1200 μF, and ∞Ω in a Gene Pulser xCell electroporation system (Biorad). 24 h post-transfection, the medium was refreshed and supplemented with 1 μg mL−1 puromycin (Sigma-Aldrich, P8833) for 36 h. Drug-resistant clones were characterized by PCR spanning the deletion. All other cell lines were generated by transfecting the plasmids with Lipofectamine 2000 (Invitrogen, 11668019) in 6 × 10⁵ mESCs dissociated into single-cells. Each transfection contained 2 μg of the homology arm vector and 1 μg of PX459 containing gRNAs targeting the 5′ or 3′ end of the deletion. 24 h post-transfection, the medium was refreshed and supplemented with 1 μg mL−1 puromycin for 48 h followed by refreshing with medium supplemented with 285 μg mL−1 Geneticin (G418)(Gibco, 11811-031) for 4–6 days. No Geneticin/neomycin selection was performed for generating the *Rnf12*$^{+/-}$ and the *Rnf12*$^{+/-}$*Pdzd4*$^{+/-}$ cell lines. Drug-resistant clones were characterized by PCR targeting genotype-specific length polymorphisms inside and outside the targeted deletion (Supplementary Table 5). CRISPR-Cas9 targeting of X-linked regions results in a population of cells losing part of the X chromosome extending beyond the targeted regions. These

deletion clones were identified and characterized by PCR (Del AB, C-G, and I-TEL). Deletion clones were identified and characterized by PCR and the deletion boundaries were determined using allele-specific RNA sequencing data. Another population of cells lost the entirety of one of the X chromosomes. These XO mESC lines were included in the bulk RNA-seq analysis as treatment-matched XO controls.

## Cell culture

Mouse embryonic fibroblasts (MEFs) were cultured in MEF medium [DMEM (Gibco, 10829-018) supplemented with 10% (v/v) fetal bovine serum (FBS, Gibco, 10500-064), 1% (v/v) penicillin/streptomycin (P/S, Gibco, 15140122), 1% (v/v) Glutamax (Gibco, 35050038), 1% (v/v) non-essential amino acids (NEAA, Gibco, 11140035), and 0.008% (v/v) beta-mercaptoethanol (Gibco, 31350010)]. Mouse iPSCs and mESCs were grown on Mitomycin-treated MEFs in mESC medium [KnockOut DMEM (Gibco, 10829-018) supplemented with 15% FBS (Gibco, 10500-064), 1% (v/v) penicillin/streptomycin (P/S, Gibco, 15140122), 1% (v/v) Glutamax (Gibco, 35050038), 1% (v/v) NEAA (Gibco, 11140035), 0.008% (v/v) beta-mercaptoethanol (Gibco, 31350010), and mouse LIF (1000 U/mL, home-made) at 37 °C under 5% $CO_2$. $Rnf12^{+/-}$, deletion lines A to K and matched controls were grown on irradiated MEFs in adjusted mESC medium [DMEM (Gibco, 11995065), 15% FBS (Capricorn Scientific, FBS-12A), 1% (v/v) penicillin/streptomycin (Sigma-Aldrich, P0781), 1% (v/v) NEAA (Lonza, BE13-114E), 0.1 mM beta-mercaptoethanol and LIF (5000 U/ml, home-made)]. mESCs and iPSCs were passaged every two days using trypsin dissociation.

## Cell sorting

Cells were dissociated using trypsin digestion and subsequently washed in incubation buffer (1× PBS, 0.5% BSA, 2 mM EDTA) and filtered through a Falcon 40 μm Cell Strainer (Corning, 352340). 1 μg SSEA1-PE antibody (R&D Systems, FAB2155P) was added per 5 million cells. Cells were incubated for 30 min at 4 °C. Cell death exclusion was performed by adding DAPI (Sigma-Aldrich, D9542) prior to sorting. Sorting was performed on a BD FACSAria Fusion (BD Biosciences) and performed by expert operators at the VIB FACS core.

## Single-cell RNA-seq library preparation

Single-cell RNA-seq libraries were prepared by following a previously published protocol (Smartseq3xpress)[44], with minor modifications. Library quality was checked using an Agilent Tapestation as well as an Agilent 2100 Bioanalyzer. Library concentration was checked with a Qubit fluorometer (Qubit dsDNA HS Assay).

## Single-cell RNA-seq sequencing

Single-cell RNA-seq libraries were sequenced in paired-end mode (150 bp) by BGI Genomics (Shenzhen, China) on a DNBSEQ-G400 sequencer (MGI Tech).

## Single-cell RNA-seq analysis

Single-cell RNA-seq data was obtained as fastq files from BGI Genomics. The zUMIs[71] pipeline was used to map reads using STAR[72] (v2.7.1a) with the options "--clip3pAdapterSeq CTGTCTCTTATACACATCT". Reads containing barcodes which corresponded to XX, XO, Deletion 1, Deletion 2, or Deletion 3 cells were mapped to an GRC38.p6 mouse reference genome N-masked for SNPs between the 129/SvJ and Cast strains and SNPs between Bl6 and 129. Reads containing barcodes which corresponded to XY cells were mapped to an GRC38.p6 mouse reference genome N-masked for SNPs between the Bl6 and Cast strains. Reads were filtered such that reads containing at least 3 bases with Phred quality score less than 20 were omitted. Allele-resolution counts were generated using a publicly available script "get_variant_overlap_CAST.R" (https://github.com/sandberg-lab/Smart-seq3/)[73] and a custom list of SNPs between 129/SvJ and Cast, with overlapping SNPs between 129/SvJ and Bl6, as well as overlapping SNPs between

Cast and Bl6 omitted. Allelic exon UMI counts were used to generate Seurat[74] (v4.2.1) objects, which were then filtered such that cells with less than 10,000 UMIs for either allele were omitted. After filtering, we obtained 69 $X^{Cast}X^{Mus}$, 44 $X^{Cast}O^{Mus}$, 6 $X^{Mus}O^{Cast}$, 61 $X^{Mus}Y$, 52 Deletion 1, 51 Deletion 2, and 41 Deletion 3 cells. Read counts were normalized to TPM. Normalized allele-resolution UMI counts were used for all analysis on the scRNA-seq dataset. Mean normalized gene expression was calculated for genes between the 10th and 90th percentile of expression per cell for each gene set of interest (all X-linked genes, inside/outside each deletion, inside/outside inactivated region), and for all (other) autosomal genes. X-to-autosome ratio was calculated by dividing the mean expression of genes in each subset by the median expression of autosomal genes.

## Characterization of X-chromosome status

The mean expression of X-linked genes from each allele was considered for each cell. If the mean allelic expression was greater than two for the *Cast* allele and less than two for the *Mus* allele, then the cell was defined as $X^{Cast}O^{Mus}$, and vice versa. One cell was omitted due to ambiguous X-status.

## Characterization of pluripotency status

The pluripotency marker genes *Essrb*, *Nanog*, and *Pou5f1* were used to assess pluripotency status. High expression levels ($\log_2(\text{TPM}) > 2.5$) of each marker gene was considered to be sufficient to conclude the cells had not differentiated and remained pluripotent.

## Identification of XCU genes

The expression values of each X-linked genes from the active allele of cells with a single active X chromosome ($X^{Cast}$ expression from $X^{Cast}O^{Mus}$, $X^{Mus}$ expression from $X^{Mus}O^{Cast}$ and $X^{Mus}Y$) was tested to be greater than half of the total allelic expression from XX cells via Student's *T*-test. Genes with $p < 0.05$ and fold change > 1.2 were defined as being transcriptionally upregulated.

## Characteristics of XCU genes

A list of known escapee genes was downloaded from Berletch et al. 2015[56] and overlapped with the list of upregulated genes in $X^{monosomy}$ and non-upregulated genes. The EMBL-EBI protein complex database[75] was used to compare the proportion of genes which code for proteins in complexes between upregulated and non-upregulated genes. The CD-CODE database was used to determine the proportion of transcriptionally upregulated X-linked genes which code for proteins that form condensates[76]. Human orthologs of upregulated X-linked genes were identified using the BioMart tool from Ensembl[77]. This list of orthologs was then overlapped with a list of known disease-causing genes from the OMIM database[78]. Significant differences between the lists of genes were calculated using a Student's *t*-test when comparing expression and a Chi-square test when comparing proportions.

The lists of upregulated and non-upregulated genes were examined for gene ontology (GO) biological processes (BP) and protein-protein interaction (PPI) enrichment using Enrichr[79]. Significant terms ($p$-value < 0.05) were selected for each database and gene list, and the top 10 terms for each category were visualized in barplots.

## Bulk RNA-seq

Cells were cultured at least two passages before harvesting. mESCs were depleted of MEFs by pre-plating dissociated cells for 45 min at 37 °C in non-gelatinized plates. The supernatant containing mESCs was spun down for 5 min at 300 × *g*. Cell pellets were lysed and RNA was extracted using the ReliaPrep™ RNA Miniprep kit (Promega, Z6012) with increased DNase treatment (25 min at room temperature). MEF depletion was validated on the genomic DNA of part of the cell pellet using PCRs targeting genotype-specific length polymorphisms and SNPs.

Sequencing libraries were prepared using the Truseq stranded mRNA library preparation method from Illumina. The libraries of the XO cells were subsequently sequenced on an Illumina NextSeq2000 sequencer resulting in single read clusters of 50 bases in length. The libraries of all other cell lines were sequenced on a HiSeq2500 sequencer, resulting in paired-end clusters of 50 bases.

## Bulk RNA-seq analysis

The SNPs in the 129/Sv and Cast/Ei lines were downloaded from the Sanger Institute (v.5 SNP142)[80]. These were used as input for SNPsplit[81] (v0.3.4), to construct an N-masked reference genome based on mm10 in which all SNPs between 129/Sv and Cast/Ei are masked. The reads were trimmed and aligned to the N-masked reference genome using TrimGalore[82] (v0.6.7) and HISAT2[83] (v2.2.1), respectively. SNPsplit was then used to assign the reads to either the 129/Sv or Cast/Ei BAM file based on the best alignment or to an unassigned BAM file if mapping to a region without allele-specific SNPs (--paired for the paired-end samples). The allele-specific and unassigned BAM files were merged into a composite BAM file using Samtools[84] (v1.10). The number of mapped reads per gene was counted for both alleles separately and for the composite BAM file using featureCounts[85] (v2.0.6) based on the gene annotation from Ensembl v98 (-t exon -s 2). The raw counts were normalized to TPM.

## Validation of bulk RNA-seq samples

All bulk RNA-seq samples were generated in a *Rnf12*[+/−] parental cell line. To assess a potential bias, we performed differential expression analysis between WT XX and *Rnf12*[+/−] XX mESCs using DESeq2[86] (v1.30.1) with default settings. Results were plotted in a scatter plot showing mean TPM in both cell lines, in which all differential expressed genes (adjusted *p*-value < 0.05, $\log_2$(fold change) < −1 or > 1) are highlighted. The deleted regions in all samples were validated by calculating the ratio of each sample to the mean of the parental *Rnf12*[+/−] samples. One deletion A clone and both deletion G clones were excluded due to a suspected recombination event rather than deletion. To ensure comparability across the remaining samples, we assessed the expression levels of the pluripotency genes *Essrb*, *Nanog*, and *Pou5f1* by plotting their TPM values for all samples. Additionally, global expression patterns were evaluated by calculating the Pearson correlation coefficients for TPMs of all autosomal genes, which were visualized in a heatmap.

## XCU in bulk RNA-seq data

To maximize statistical power, we analyzed non-allelic TPMs of all X-linked genes within and outside each deletion region. Only expressed genes were selected by filtering for TPM > 5 in *Rnf12*[+/−] samples. Deletion I was excluded from the plots showing expression within each deletion region due to insufficient expressed genes. For each genomic region, mean TPM values were calculated. These levels were compared to the mean XX level (from *Rnf12*[+/−]) and 50% of the mean XX level, representing the expected expression level for a single X chromosome without upregulation.

To account for copy number differences, gene expression was normalized to copy number-corrected values by dividing TPM values by the number of copies present, based on the genomic locations of each deletion. Mean copy number-corrected TPMs were then calculated for genes within and outside deletion regions and compared to the mean copy number-corrected TPM in XX samples.

To validate allelic upregulation detected in non-allelic TPM values, we selected X-linked genes with a fold change > 1.2 in copy number-corrected TPMs between XO and XX cells. For this subset of upregulated genes, allelic *Cast* expression data was filtered, and the $\log_2$ fold change between XO and XX cells was calculated. The allelic upregulation status in XO cells for each gene, annotated by overlapping deletion region, was visualized in a heatmap. The level of upregulation

in the deletion samples was assessed by analyzing the copy number-corrected expression of genes upregulated in XO cells. For each gene, the $\log_2$ fold change relative to the mean expression in XX cells was calculated. These values were plotted separately for genes located inside and outside each deletion, with median values per sample indicated. For visualization, extreme values were clipped at −1 and 4. Statistical significance was assessed using two-sided paired t-tests on the mean expression per genotype, with Bonferroni correction for multiple comparisons. No statistical testing was performed for genes located within the deletions due to the limited number of genes per deletion region.

## Autosomal upregulation analysis

We reanalyzed a bulk RNA-seq dataset from GSE119602 using the same approach as described in "Bulk RNA-seq analysis". Inactive genes were defined as genes whose fold change was less than 0.15 when comparing samples with doxycycline to samples without doxycycline. The inactive region was defined as the region from 80 Mb to the 3′ telomere. Upregulated genes were defined as genes with a fold change greater than 0.5 when comparing samples with doxycycline to samples without doxycycline. Characteristics of the upregulated genes were evaluated as described in "Characteristics of XCU genes". Mean normalized gene expression within/outside the inactive region for each allele was calculated using genes with expression between the 5th and 95th percentile for each sample.

## RT-qPCR

mESCs were pre-plated as described. Cell pellets were lysed and RNA was extracted using the ReliaPrep™ RNA Miniprep kit (Promega, Z6012) with increased DNase treatment (25 min at room temperature). 500 ng RNA was reverse transcribed using Superscript III (Invitrogen, 18080093) and random hexamers (Invitrogen, N8080127) with the addition of RNaseOUT (Invitrogen, 10777019). cDNAs were diluted 1:1 in water. RT-qPCRs were performed in triplicate in the GoTaq qPCR Master Mix (Promega, A6002) in a CFX384 Touch Real-Time PCR Detection System (BioRad, LJB22YE8Z). *RplpO* and *H2afz* were used as normalization controls. Allele-specific primer pairs were optimized using pure gDNA of 129/Sv and Cast/Ei mice. All primers are included in (Supplementary Table 6).

## Proteomics

Proteomics experiments were performed on $X^{Cast}X^{Mus}$ (*Rnf12*[+/−]), $X^{Mus}Y$ (F1 2-3), and $X^{Cast}O^{Mus}$ (*Rnf12*[+]) ESCs. Additionally, two deletion cell lines (Deletion E and K in *Rnf12*[+/−]) were included, selected based on their single-region deletions and a high proportion of transcriptionally upregulated genes.

For global proteome analysis of whole cell extracts, cells were lysed in 100 mM Tris/HCl, pH 8.2, containing 1% sodium deoxycholate (SDC) using sonication in a Bioruptor Pico (Diagenode). Protein concentrations were measured using the BCA assay (ThermoFisher Scientific). 100 μg protein was reduced in lysis buffer with 5 mM dithiothreitol and alkylated with 10 mM iodoacetamide. Next, proteins were digested with 2.5 μg trypsin (1:40 enzyme:substrate ratio) overnight at 37 °C. After digestion, peptides were acidified with trifluoroacetic acid (TFA) to a final concentration of 0.5% and centrifuged at $10,000 \times g$ for 10 min to spin down the precipitated SDC. Peptides in the supernatant were desalted on a 50 mg C18 Sep-Pak Vac cartridge (Waters). After washing the cartridge with 0.1% TFA, peptides were eluted with 50% acetonitrile and dried in a Speedvac centrifuge. Peptides were then analyzed by nanoflow LC-MS/MS as described below.

Nanoflow LC-MS/MS was performed on an Vanquish Neo LC system coupled to an Orbitrap Exploris 480 mass spectrometer (both ThermoFisher Scientific), operating in positive mode and equipped with a nanospray source. Peptide mixtures were trapped on a PepMap trapping column (2 cm × 100 μm, Thermo, 164750) at a flow rate of 1 μl/

min. Peptide separation was performed on ReproSil C18 reversed phase column (Dr Maisch GmbH; column dimensions 25 cm × 75 μm, packed in-house) using a linear gradient from 0 to 80% B (A = 0.1% FA; B = 80% (v/v) AcN, 0.1 % FA) in 120 min and at a constant flow rate of 250 nl/min. The column eluent was directly sprayed into the ESI source of the mass spectrometer.

For data independent acquisition (DIA), all spectra were recorded at a resolution of 120,000 for full scans in the scan range from 350 to 1100 m/z. The maximum injection time was set to 50 ms (AGC target: 4E5). For MS2 acquisition, the mass range was set to 336–1391 m/z with variable isolation windows ranging from 7 to 82 m/z with a window overlap of 1 m/z. The Orbitrap resolution for MS2 scans was set to 30,000. The maximum injection time was at 54 ms (AGC target: 5E4; normalized AGC target: 100%).

## Proteomics analysis

DIA raw data files were analyzed with the Spectronaut Pulsar X software package (version 17.0.221202, www.biognosys.com), using directDIA for DIA analysis, including MaxLFQ as the LFQ method and Spectronaut's IDPicker algorithm for protein inference. The $Q$-value cutoff at precursor and protein level was set to 0.01. All imputation of missing values was disabled.

Proteins detected at least once in each sample were retained for further analysis. The mean abundance of all X-linked proteins in each XX, XO, and XY sample was plotted with confidence intervals using seaborn pointplot. To avoid a detection bias of low-abundance proteins between X-linked (1n in XO and XY cells) and autosomal proteins (2n), X:A ratios were calculated using only proteins with a mean abundance greater than 12,000,000 in the XX samples (2n for all proteins). The X:A ratio was defined as the abundance of each X-linked protein divided by the mean autosomal abundance in the same sample, and visualized using seaborn pointplot.

Since proteomics data lacks allele-specific resolution, we corrected the data for the copy number to identify significantly upregulated proteins. The abundances in XX samples were divided by 2 to account for two alleles, while abundances of XO and XY samples remained unchanged. Moreover, autosomal abundances were divided by 2 in all samples. Differential expression analyses were performed between XO and XX and between XY and XX using a Tuckey's HSD test. Upregulated proteins were selected based on the criteria of log₂(fold change) > 0 and $p$-value < 0.05 in both XY and XO comparisons. The results for all X-linked proteins were visualized using volcano plots, and the number of upregulated proteins per chromosome were displayed in a barplot. To assess the overlap in upregulation between XO and XY cells, gene lists identified in each analysis were compared using a Venn diagram. In addition, a scatter plot was generated to visualize the correlation between cell lines, showing the log₂ fold change in copy number-corrected abundances for XO vs XX and XY vs XX. The correlation between transcriptomic and proteomic upregulation was explored using a scatter plot showing the log₂ fold change of the copy number-corrected abundances in XO versus XX and the copy number-corrected transcript expression in XO versus XX.

Samples with deletions E and K were analyzed by labeling proteins according to the genomic location of the gene encoding the protein (inside or outside the deleted region). For each region separately, the mean normalized protein abundance was plotted, comparing levels outside and inside the deletion to the XX reference level and 50% of the XX reference level, respectively. Abundances were copy number-corrected by halving abundances for two-copy proteins. These corrected abundances were visualized in a heatmap, focusing on proteins significantly upregulated in XO and XY samples.

## ChIP-seq

mESCs were depleted of MEFs 16 h before the harvest by pre-plating dissociated cells for 45 min at 37 °C in non-gelatinized plates. In total,

$1 \times 10^8$ cells were crosslinked by the addition of 1% formaldehyde (Sigma-Aldrich, F1635-25ML) to the dish and incubated at room temperature for 10 min on a rocking platform. Formaldehyde was quenched by adding 125 mM glycine for 5 min. Subsequently, cells were washed once with 1 × PBS before scraping in 1 × PBS supplemented with 1× protease inhibitors (Roche, 11836170001). Cells were pelleted, resuspended and incubated for 10 min at 4 °C in 1 × LB1 (50 mM HEPES pH 8.0, 140 mM NaCL, 1 mM EDTA, 10% glycerol, 0.5% NP-40, 0.25% Triton X−100, 1× protease inhibitors), in 1 × LB2 (10 mM Tris-HCl pH 8.0, 200 mM NaCl, 1 mM EDTA, 0.5 mM EGTA, 1× protease inhibitors) and in 1× LB3 (10 mM Tris-HCl pH 8.0, 100 mM NaCl, 1 mM EDTA, 0.5 mM EGTA, 0.1% SDC, 0.5% N-laurolylsarcosine, 1× protease inhibitors). DNA was sonicated to an average fragment length of 100−300 bp using a Bioruptor Pico (Diagenode, B01060001) for 10 cycles of 30 s on and 30 s off. 1/20th volume of 20% Triton X−100 was added to the sonicated chromatin, and incubated for 10 min at 4 °C before centrifugation for 10 min at 4500 × g 10 at 4 °C to remove debris. 10 μl chromatin was used to measure the chromatin concentration by nanodrop and check the sonication efficiency. Sheared chromatin was flash frozen in liquid nitrogen and stored at −80 °C until further use. For H4K16ac ChIPs, 30 μg of chromatin was diluted to 500 μl in ChIP dilution buffer (16.7 mM Tris-HCl pH 8.0, 167 mM NaCl, 1.2 mM EDTA, 1.1% Triton X−100, 0.01% SDS, 1× protease inhibitors) and incubated while rotating overnight at 4 °C with 2 μg H4K16ac antibody (Sigma-Aldrich, 07-329, lot: 3772263) or IgG (Sigma-Aldrich, 12-370). 30 μl of Magnetic beads (Life Technologies, 10004D) were washed once in ChIP dilution buffer, added to each sample and incubated while rotating for 4 h at 4 °C. Beads were washed 3× in Low salt buffer (20 mM Tris-HCl pH 8.0, 150 mM NaCl, 2 mM EDTA, 1% Triton X−100, 0.1% SDS, 1× protease inhibitors), 1× in High salt buffer (20 mM Tris-HCl pH 8.0, 500 mM NaCl, 2 mM EDTA, 1% Triton X−100, 0.1% SDS, 1× protease inhibitors), 1× in LiCl buffer (10 mM Tris-HCl pH 8.0, 250 mM LiCl, 1 mM EDTA, 1% NP-40, 1% SDC, 1× protease inhibitors) and rinsed once in TE (10 mM Tris-HCl pH 8.0, 1 mM EDTA), 10 min rotating at 4 °C each wash. Chromatin was eluted in 200 μL Elution Buffer (1% SDS, 0.1 M NaHCO3) with 5 μL protease K (10 mg mL⁻¹) and 2 μL RNAse A (10 mg mL⁻¹) while shaking at 1000 rpm on an orbital thermal-mixer for 2 h at 37 °C followed by 65 °C overnight. DNA was purified using the Zymo ChIP DNA clean concentrator kit (Zymo Research, D5202) according to the manufacturer's instructions and eluted in 15 μL H2O. 0.1 μL DNA was used per qPCR to validate ChIP enrichment.

The ChIP-seq libraries were prepared with the ChIPThruPLEX method (Takara Bio, R400675). These libraries were subsequently sequenced on an Illumina NextSeq2000 sequencer. Paired-end clusters were generated of 50 bases in length.

## ChIP-seq analysis

The ChIP-seq samples were analyzed allele-specifically, similar to the bulk RNA-seq analysis. The reads were trimmed using Trim Galore and mapped to the N-masked reference genome using Bowtie2[87] (v2.5.0). Bam files were split by allele using SNPsplit and the resulting allele-specific bam files were normalized to counts per ten million (CP10M) using the bamCoverage tool (-bs 1 --normalizeUsing CPM) from deepTools[88] (v3.5.5). Moreover, allele-specific tracks from GSE116480 were downloaded, including CP10M-normalized data for H3K17me3, H3K27ac, H4ac, H2AK119Ub, H3K4me1, H3K4me3, and H3K9ac after 0, 4, 12, and 24 h of doxycycline.

Enrichment of all ChIP-seq tracks was plotted for each allele separately on the genomic regions of upregulated and not upregulated genes, selected based on the scRNA-seq analysis. Mean enrichment per bin was calculated using deepTools computeMatrix (scale-regions -bs 100 --averageTypeBins mean --missingDataAsZero -m 3000 --upstream 1500 --downstream 1500). Profile plots were generated using custom Python (v3.9.16) scripts. The matrix file from deepTools was loaded,

and average enrichment profiles per bin were plotted using seaborn line plot, including standard error bands.

To evaluate temporal patterns and assess statistical significance, we calculated the mean signal in two genomic regions: (1) the promoter region, defined as 1 kb upstream of the TSS to the TSS, and (2) the gene body, defined as the region spanning from TSS to TES. For each region, bed files were generated containing the corresponding genomic coordinates for all genes. These bed files were used as input for deepTools multiBigwigSummary, along with all allele-specific bigwig files, to calculate the average scores in these regions. Changes over time were visualized with line plots, showing the mean CP10M score with standard errors for upregulated and not upregulated genes separately.

Statistical testing was performed for each gene set separately. Significant differences between 0 and 24 h were calculated using two-sided paired $t$-tests, with $P$-values corrected for multiple testing with the Benjamini–Hochberg approach. These adjusted $P$-values were added next to each gene set in the ChIP-seq line plots. Additionally, differences between upregulated and not upregulated genes were tested using two-sided Student's $t$-tests, also corrected for multiple testing. The resulting significance was added above the corresponding regions in the profile plots.

Genome browser overviews were created to show ChIP-seq enrichment on the active upregulated allele at the loci of three upregulated genes. Tracks were selected based on significant differences in enrichment between 0 and 24 h, as well as between upregulated and non-upregulated genes at 24 h. CP10M-normalized allele-specific tracks for the significantly enriched marks were visualized using IGV v2.19.4[89].

## Statistics and reproducibility
No statistical method was used to predetermine sample size. One cell was excluded from the scRNA-seq analysis due to ambiguous allelic expression from the X chromosome. One deletion A clone and both deletion G clones were excluded from RNA-seq analysis due to suspected recombination events or partial XO genotype, rather than the intended deletion.

## Reporting summary
Further information on research design is available in the Nature Portfolio Reporting Summary linked to this article.

## Data availability
The RNA-seq and ChIP-seq data generated in this study have been deposited in the NCBI Gene Expression Omnibus (GEO) database under accession code GSE282792. Moreover, RNA-seq and ChIP-seq datasets from GSE119602 and GSE116480 were reanalyzed. The mass spectrometry proteomics data have been deposited to the ProteomeXchange Consortium via the PRIDE partner repository with the dataset identifier PXD057399. Source data are provided with this paper.

## Code availability
Code is available on the Pasque Lab GitHub account and can be accessed via the following link: https://doi.org/10.5281/zenodo.16760254[90].

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

## Acknowledgements

We thank Cheryl Maduro and Büşra Göynük for technical support in generating the cell lines, the VIB FACS Core for their assistance in sorting cells, the KU Leuven Genomics Core for assistance with sequencing, as well as the Vlaams Supercomputer Centrum for their computational resources. We thank Björn Reinius and our colleagues for many helpful discussions. The V.P. laboratory is supported by the Research Foundation-Flanders (FWO grants G0C9320N and G0B4420N to V.P.), KU Leuven Research Fund (C1 grant C14/21/119 to V.P.), and Pandarome project 40007487 (G0I7822N) (funded by the FWO and F.R.S.-FNRS) under the Excellence of Science (EOS) program. R.N.A. is supported by FWO fellowship (11L0724N). J.B. is supported by the Oncode Institute. S.M. is supported by ZonMW-Top subsidy (40008129815046).

## Author contributions

Conceptualization: V.P., R.N.A., J.G., B.F.T, J.B., H.M.B., C.G. Data curation: R.N.A., B.F.T., J.B., S.M. Formal analysis: R.N.A., B.F.T., J.B., S.P., W.F.J.I, J.A.A.D. Funding acquisition: V.P., J.G. Investigation: V.P., R.N.A., J.G., B.F.T, J.B., H.M.B., C.G. Methodology: R.N.A., B.F.T., J.B., S.P., W.F.J.I, J.A.A.D. Project administration: V.P., J.G. Resources: V.P., J.G., J.A.A.D. Software: V.P., J.G., J.A.A.D. Supervision: V.P., J.G., H.M.B., C.G. Validation: R.N.A., B.F.T., J.B. Visualization: R.N.A., J.B., B.F.T. Writing, reviewing, and editing of the manuscript: V.P., R.N.A., J.G., B.F.T., J.B., H.M.B., C.G.

## Competing interests

KU Leuven in Belgium has filed an international patent application (PCT/EP2023/073949) for methods to create ExM cells from naive human PSCs. V.P. is one of the inventors of this patent, together with Amitesh Panda, Thi Xuan Ai Pham, and Bradley Balaton. The patent is currently pending. The patent does not cover a specific aspect of this manuscript. The remaining authors declare no competing interests.
