## [Peer Review file · Nature Communications]

X-chromosome upregulation operates on a gene-by-gene basis at RNA and protein levels

Corresponding Author: Professor Vincent Pasque

Version 0:

Reviewer comments:

Reviewer #1

(Remarks to the Author)

In this exciting manuscript, the authors report that mouse embryonic stem cells compensate for expression levels in the context of heterozygous deletions on the X chromosome and heterozygous “knock-downs” on autosomes. Using mESC lines from hybrid crosses, therefore allowing allele-specific analyses, the authors first show that mESC with a single X chromosome (XO or XY) display X-chromosome upregulation (XCU), confirming previous reports; in other words, expression from the single X chromosome is approximately double of that of each X chromosome in XX cells. The authors show that XCU is associated with gain of active chromatin marks H3K9ac and H3K4me3, but not H4K16ac, which is associated with the well-known dosage compensation mechanism via XCU in fruit flies. Whether the histone modifications are a cause or a consequence of XCU in mESC remains an open question. The authors then profiled a series of XX mESC clones harbouring large heterozygous deletions collectively covering the entire X chromosome and showed that (1) no deletion leads to global XCU on the other X chromosome, excluding the existence of a potential “X-upregulation centre” (akin to the “X-inactivation centre” for X-chromosome inactivation); (2) genes within the monosomic region are upregulated; (3) remaining genes in trans are not upregulated. Thanks to the use of both transcriptomic and proteomic approaches, the authors show that upregulation of the monosomic region in the context of the heterozygous deletions, or of the X chromosome in the context of XY and XO cells, happens at both RNA and protein level. Interestingly, and as previously suggested, the authors find that global XCU occurs on a gene-by-gene basis and involves post-transcriptional mechanisms, since for some genes dosage compensation is observed only at the protein level.

Overall, this study is of high quality and provides unexpected results and important insights into how dosage compensation operates in mammals, using a systematic series of chromosomal segment losses along the X chromosome. We would like to recommend this manuscript for publication, however a number of points should be addressed, especially regarding the definition of “upregulated” genes, which has consequences for several of the downstream analyses. While we believe the conclusions will most likely remain unaffected, the analyses might have to be revisited if a new set of “upregulated” genes is defined.

1/ The authors found that, for genes for which they obtained allele-resolution data from single-cell RNA-sequencing, 40% were significantly upregulated in XO and XY cells compared to XX. Based on Extended Data Fig. 1C, this number seems to be calculated solely on the basis of statistical significance, ignoring the actual fold change. In other words, for some genes the upregulation seems to be very close to zero, yet statistically significant. Do the authors consider that this has biological meaning? Inversely, there are genes for which the fold change is quite considerable but statistical significance is not reached. Maybe the authors could present a more nuanced view with numbers for the different scenarios (40% when considering only statistical significance, X% genes above fold-change bigger than Z, Y% genes when considering both fold-change and statistical significance). How the “upregulated” genes are defined has an impact on subsequent analyses, so it would make sense to rerun the analyses (at least some) with the alternative set of “upregulated” genes.

2/ Why do the authors state that the occurrence of XCU at the protein level is not clear (lines 333-334)? To our knowledge, there is only one study addressing this, which the authors properly cite, and this study concludes that there is compensation at the protein level. Do the authors have reason to believe that the conclusion is disputable? This could be stated if so.

3/ It would be helpful to acknowledge in the text that for sample X(Mus)O the number of single cells analysed was much lower than for other samples.

4/ Can the authors show the plots in Fig. 1B-C but only for the upregulated genes?

5/ The authors state that their findings regarding the histone modifications “overturn the view that these are not enriched on the Xu” (line 202). Could the authors provide a reference for this view? Other studies have associated XCU with specific histone modifications (including a study from the authors).

6/ The authors state that “both species appear to have evolved the use of different histone marks for XCU, H4K16ac in *Drosophila*, H3K9ac and H3K4me3 in mice”. This seems to imply that hPTM are instructive for XCU in mice, which has not really been shown? They might just be a by-product of increased transcriptional activity? Yet, this might not totally be the case either, since the authors show increased hPTM over time even for genes that are not upregulated, suggesting that hPTM can precede increased gene expression; something that the authors do not mention but could be worth discussing. Still, these data would suggest that hPTM are not sufficient to induce XCU, and therefore not instructive (unlike the scenario in *Drosophila*).

7/ The XCU centre hypothesis needs to be further and explicitly explained (and would it make sense to call it “Xuc”, analogous to the “Xic”?). This explanation needs to include a definition of the putative “Xuc”, how many would be necessary, how it would work (in cis, in trans) and what would be expected upon its deletion. The schemes provided in Fig. 2B and 2D are not self-explanatory, it is hard to understand what the authors wanted to convey.

8/ The authors use XCU to refer to either what is observed at the level of single genes or at the level of averaged expression levels from an entire chromosome; this sometimes leads to confusion while reading. For example, in lines 243-244, the authors state that “none of the three deletions examined induced XCU across the X in trans”; we suggest that the authors refer to “global XCU” to avoid confusion (the subtlety of “across the X” might be missed), since the deletions do induce XCU in trans (for genes within the deleted region). Likewise, the authors might also use “full XCU” vs “partial XCU” in specific instances. For example, in lines 268-269, the authors state that “none of the 14 new deletions induced XCU for genes located in trans outside of the deleted regions (Figure 2J)” – the authors might consider adding “full” to XCU, given that for some deletions there seems to be some upregulation (deletions C-G and F). Additionally, would it make sense to run statistics on these data?

9/ When analysing the compensation for genes inside the deletions, in particular Fig. 3E, the authors make no comments or mention to the fact that: (1) for some regions (ITEL and J), there is no XCU (ITEL) or even a decrease in expression levels (J) in XO compared to XX cells; are the changes observed upon deletion related to the data in Fig. 3G? (2) for some deletions (D, K), XCU is not “full” but “partial” (see previous point too). Is this partial XCU for deletions D and K linked to the results shown in Extended Data Fig. 4F? Why do the authors think these deletions are behaving differently?

10/ Can the authors show or state an overall quantification of Fig. 3F and 3G? What is the “large majority” (line 289)?

11/ The authors asked whether genes upregulated in XO cells were also upregulated when deleted in cells with large heterozygous deletions. Could the authors also do the reciprocal analysis, i.e., looking at whether genes upregulated in the deletions are upregulated in XO? Or is it redundant given a high overlap? This should be mentioned in either case.

12/ Within the deleted regions, how many genes get upregulated? The authors could also provide a table (or equivalent) showing how many genes are contained in each of their deletions (both the ones named with numbers and the ones named with letters).

13/ In Fig. 3F/3G, why are some genes in grey for the XO if the starting point for the analysis is genes that are upregulated in the XO? We thought that only upregulated genes were being shown, but maybe the authors are showing all genes within the deletion? It would be helpful to have the names of the genes below.

14/ For overlapping deletions (e.g., deletion 1 and deletion B), is there concordance in which genes are upregulated?

15/ Can the authors add to the scheme in Fig. 2H or 2I how the deletions 1, 2 and 3 overlap with the others? This is represented in Extended Data Fig. 4B, but it would be important to have it in the main figures anyway.

16/ In Fig. 3H, the authors should show the results for the parental line of the clones, Rnf12+/- Pdzd4+/?

17/ In Extended Data Fig. 2C-E, it is hard to appreciate big differences. Could statistical testing help clarify?

18/ The authors noted “a slight increase in gene expression in trans in two cell lines, Deletion 1 and 3, but the magnitude of the increase was low” (Lines 244-246). Is this explained by the upregulation observed for genes in trans for the deleted region?

19/ When discussing the results in the autosomal context (relative to Fig. 4), the authors often refer to genes being “silenced” or “inactivated”, but a more accurate description would be that they see reduced gene expression levels (at the bulk level)? Can we really talk about silenced/inactivated genes without single-cell data?

20/ In Fig. 4 (especially C and E), wouldn't it make sense to plot the data only for the genes that are affected by Xist expression, instead of the entire 80-160Mb region?

21/ How did the authors select the “deletion” lines to analyse by proteomics? In other words, why deletions E and K? Were there specific reasons?

22/ A number of sentences need polishing / corrections:

- Lines 86-88 (sentence is not coherent)
- Line 139 (“in cells with XO and XY cells”)
- Lines 144-148 (“which” vs “that”)
- Lines 170-173 (rephrase for simplicity, two sentences could be merged into one)
- Line 257 (assuming the authors mean “syntenic” and not “synthetic”)
- Line 266-267: the results the authors refer to are not shown in Fig. 2J
- Lines 289–290: slightly confusing — the authors mention that the effect is not happening outside of the deletion region but the cited figures only show data for inside the deletion
- Line 347–349: the sentence is unclear and should be simplified
- Extended Figure 6A: typo in y-axis label

Reviewer #2

(Remarks to the Author)

Reviewer #3

(Remarks to the Author)

The manuscript addresses the interesting question of whether X chromosome upregulation is controlled via a single regulatory centre (akin to the X inactivation centre), or rather on gene-by-gene and protein-by-protein level. They use pluripotent mouse cells (ES cells and iPSC cells) with differing X chromosome complements, along with heterozygous X chromosome fragment deletions in combination with single cell RNA-seq and proteomics to tackle their question. Their data support a gene-by-gene model of X chromosome upregulation that can occur at an RNA level and/or a protein level, frequently both. While upregulation at the RNA level has been reported by several papers, the authors' novel findings relate to tackling the question of whether it is controlled chromosome-wide or not. Interestingly, they also report that upregulation can occur on autosomes (though to a smaller extent), which was previously known. They show it can occur as a function of silencing of one allele rather than only genetic disruption as occurs in cancer. This section is less well explored. Overall, the paper was enjoyable to read, adds to the existing literature on X chromosome upregulation (which has been a controversial topic, so additional well-controlled studies are valuable), and I think will be of interest to the readers of Nature Communications. The major result that dosage compensation is regulated at a gene-by-gene level has major implications for interpretation of disease and potentially also disease treatment and therefore is likely to be relevant beyond the X chromosome regulation field.

There are several areas in the text, figure legends and methods that need clarification and potentially some limited further experiments. My suggested changes aim to provide confidence in the data or to provide clarity for the reasoning and interpretation for readers who aren't entrenched in the X chromosome dosage compensation field.

XX, XO and XiXa status of mESC and iPSC systems:

It isn't clear why iPSC are used in some sections and ESC in others, and a mix of two are compared. The authors need to provide explanation for their choice, and then also show that there is no difference between the 2 for XCU upregulation purposes, so that the data can be appropriately interpreted. For example, for Figure 1, only the XY samples are ESCs, so it needs to be clear that comparing male to XO and XX iPSC is reasonable. For Figure 2, deletions are made in XX ESC, but there are no control XX ESCs without deletions (presumably the parental line), only iPSC. These controls in each case are essential for appropriate interpretations, particularly when there are very few single cells from XO iPSC in Figure 1 and so there is a greater reliance on XY ESC data.

Female iPSC and ESC can lose an X chromosome and become XO (which is useful in this study). In serum + LIF culture systems as used here, the cells can also undergo differentiation and become XiXa. The methods and text do not indicate how the XX populations (mESC or iPSC) were checked for their XX vs XO status, and X inactivation status. These details are also critical and need to be included.

RNAseq data presentation:

The cell-to-cell variability of mean gene expression (summarised for X-linked genes or subsets) is clearly presented in the scRNAseq data. It is also important to see the range of expression across the genes in the plot for the bulk RNA-seq data, by plotting all genes as separate data points, with replicates side by side. This would give a better sense of variability, which is important when sometimes the effects being interpreted are very small e.g. Fig. 1J, Fig. 3E

XCU hypothesis and histone modifications:

The section of 'No evidence for an XCU center' does not lay out the hypothesis that there might be an XCU, akin to a XIC for inactivation. While it is mentioned briefly in the introduction, a reminder in the first sentence of this section and also relating it to the XIC would help to set the scene for the experiments.

The statistical analyses on Figure 1I/J are not immediately easy to read with regards to the timecourse of ChIPseq. Which asterisks refer to which analyses? Can the authors clarify if it was only 0 and 24 hours used for the timecourse analysis, as indicated? If so, why weren't all data points used?

While the main data is about the 'active' histone marks, the polycomb marks were analysed similarly, using just the promoter marking. Here an analysis using the enrichment over the whole gene and the upstream region would be better. In addition, can some example plots be included in supp figures of the appearance of different ChIPseq data tracks to allow readers to assess the data for themselves.

Line 170, no enrichment of genes encoding proteins that are in condensates. Given the challenges studying condensates, and the relatively new area of research, a caveat here that not all proteins that undergo condensation may have been identified seems in order.

Line 171. The comment that XCI escapees aren't upregulated needs explanation. Readers that aren't XCI aficionados will need an explanation as to why these would have been tested.

Deletion interpretations:

Line 280/81, Fig 3E. The authors state that similar observations to the 3 large deletions were made with 12 smaller deletions. However, there appear to be differences in some e.g. C, D, I-TEL, J and K. There appears to be no effect in some e.g. C, half the level of upregulation as in monosomy X in others e.g. K, and lack of an effect in the XO in others again (I-TEL, K). So more explanation and discussion is required here, particularly as the results don't easily tally with the data shown in Fig. 3G.

Here it is not clear why these deletions were made in Rnf12^{+/-} cells. This needs explanation. Was there an advantage, or were they just the cells that were around? Was the idea to know which allele had the deletion in some way?

Autosomal Xist interpretation:

Fig. 4B is summarised data, but isn't showing downregulation of the genes. Can the authors include a data plot so show the data for silencing?

Here the authors should discuss why the level of upregulation is much smaller than on the X, and also explain their interpretation. On line 321/322 they say the upregulated genes are not higher, in fact they are statistically LOWER in expression than those that aren't upregulated. This should be made clear.

They also say there is no significant difference in genes encoding protein complexes, but the difference is larger than those shown for the XCU data but is apparently not significant. This doesn't quite make sense and should be checked. It would be FEWER genes that are part of complexes that are upregulated, contrasting with the XCU data.

These data together suggest that any upregulation observed, which seems smaller in effect size, is perhaps less reliable than the XCU. Could this be because the cells don't tolerate chromosome 3 silencing well, limiting the capacity to see upregulation in trans?

Protein-level compensation:

It isn't clear what the overlap is between XY and XO differential proteomic analyses. I also could not find if all samples were ESC here, or a mix of ESC and iPSC. Can the authors include a comparison between XY and XO to give a high confidence list of XCU proteins? If the list is as different as it appears on the Figure, what is the explanation? The labels on Figure 5 of proteins don't allow such a comparison.

Can the authors explain the X:A ratio analysis in Ext Fig 6B – why would XX cells have X:A ratio of >1?

Figure 5F shows 8% of proteins that have RNA upregulation but not protein level. The confidence in this group is presumably low, given proteomics doesn't have the resolution of transcriptomics, and this should be mentioned. It doesn't make a lot of evolutionary sense to have RNA but not protein level upregulation, as potentially a wasted effort in production. It isn't clear why E and K deletions were chosen for proteomic analyses, especially when K doesn't seem to have 'standard' behaviour at the RNA level. Was this the reason why? In addition the XO and XY do not seem to be similar, but this is not explained either by ESC vs iPSC if this is the case, or another reason. This needs to be explained.

The description of this final paragraph of the results is fairly repetitive and needs to be improved/

Methodological details:

Detailed methods for culture of the mESC and iPSC on MEF feeder cells are not provided. More details are required.

The methods for creation of the deletion lines are hard to follow and additional explanation of why the specific approach was taken would help e.g. why were Rnf12^{+/-} cells used as the starting point? How was the neo cassette later deleted? Why was puromycin mentioned?

Small text changes:

Line 267, Figure 2J. These data do not show uncorrected expression so don't show that there is higher X-linked expression in XX vs XO cells.

Paragraph starting at line 114, refers to results with interpretations embedded prematurely before all data is shown, so should be edited. For example, line 116 says 'As expected, we confirmed the presence of XCU when only one active X chromosome is present,' and similar interpretations throughout. At this stage it is likely more appropriate to say that higher expression was observed in cells with one X rather than 2. Then after Fig 1D/E data is presented with the X:A ratio, it can be interpreted as XCU.

Line 139 delete 'in cells with'

Line 190. Native ChIP-seq is stated, but methods mention formaldehyde fixation, so probably native should be removed.

Line 203. The authors claim some active marks are more highly enriched on genes that become upregulated during XCU, but they don't say which these are in the text, which would help interpret the figure.

(Remarks to the Author)

In this study, Allsop and colleagues investigate the mechanisms of gene dosage compensation, particularly X-chromosome upregulation (XCU), using allele- and molecule-resolution transcriptomics in mouse pluripotent stem cells. They show that segmental deletions on one X chromosome can induce gene-specific upregulation in trans, but not global XCU, and that this compensation occurs at both the mRNA and protein levels. The authors further demonstrate that specific active histone modifications (e.g., H3K9ac, H3K4me3) are enriched during XCU. Notably, they also reveal that autosomal dosage compensation can occur via transcriptional upregulation in trans following targeted inactivation in cis, which is a particularly intriguing finding. The data are comprehensive and clearly presented, offering important insights into how mammalian cells sense and respond to gene dosage imbalance through transcriptional compensation. However, several key issues need to be addressed. The specific comments are detailed below.

Major Comments:

1. Expression level of the Mus allele:

-In Figure 1b, the expression level of the Mus allele in XMusOCast appears lower than in XMusY. Is this difference statistically significant?

-How do the authors interpret this discrepancy?

-A similar concern arises in Figure 3d, can the authors clarify whether this variation is biologically meaningful or due to technical variability?

2. In Figure 1i, the authors report an increase in H2AK119Ub on the active X chromosome (Xa). Given that H2AK119Ub is typically associated with gene repression, how should this observation be interpreted in the context of XCU?

3. Autosomal compensation:

-The autosome-based findings are novel but less deeply explored. Is autosomal compensation a general phenomenon, or limited to specific loci or cell states?

-In Figure 4e, the upregulation of the Mus allele in the 80–160 Mb region does not appear to reach the level typically observed in canonical XCU. This may be due to insufficient silencing in cis, as suggested by Figure 4c. The authors may consider generating a heterozygous autosomal knockout model to more definitively test whether dosage compensation via transcriptional upregulation in trans can be induced under stronger silencing conditions.

4. Mechanistic insight into the sensing process:

While the study shows that cells can detect segmental deletions and compensate in trans, the underlying molecular sensors or signaling pathways responsible for dosage sensing remain unclear. A discussion or hypothesis about what might constitute the sensing mechanism would strengthen the manuscript.

5. Whether similar dosage compensation responses are observed or expected in human stem cells or development? The discussion could benefit from a short discussing about this.

Minor Comments:

1. Clearly defined “in trans” and “in cis” early in the introduction for general readers.

2. In Figure 2j, the TPM values for the same cell type vary across different deletion conditions. Did the authors use the same XMusXCast and XMusOCast controls in each of the deletion comparisons?

3. Additionally, in certain cases (e.g., deletion C–G) in Figure 2j, there appears to be a slight induction of XCU. Could the authors provide clarification on whether this represents a meaningful transcriptional compensation or falls within expected noise?

Reviewer #5

(Remarks to the Author)

Version 1:

Reviewer comments:

Reviewer #1

(Remarks to the Author)

The authors have fully addressed all my comments. This is a very valuable manuscript for the field of X-chromosome regulation, sex chromosomes and dosage compensation.

Reviewer #2

(Remarks to the Author)

Reviewer #3

(Remarks to the Author)

The authors have dealt with all my requests and edited the manuscript accordingly. Thank you.

I have just one clarification, related to point 2 of my review, i.e. that there could also be XaXi cells in their culture system. They have done a good job of explaining why the cells aren't differentiated, but that doesn't necessarily mean they didn't become XaXi. Did they check cells did not have Xist as well as two X chromosomes? If yes, that should be included as well.

Reviewer #4

(Remarks to the Author)

The authors have addressed my comments. Congratulations.

Reviewer #5

(Remarks to the Author)

Point by point response to the comments of the reviewers.

Reviewer #1 (Remarks to the Author):

In this exciting manuscript, the authors report that mouse embryonic stem cells compensate for expression levels in the context of heterozygous deletions on the X chromosome and heterozygous “knock-downs” on autosomes. Using mESC lines from hybrid crosses, therefore allowing allele-specific analyses, the authors first show that mESC with a single X chromosome (XO or XY) display X-chromosome upregulation (XCU), confirming previous reports; in other words, expression from the single X chromosome is approximately double of that of each X chromosome in XX cells. The authors show that XCU is associated with gain of active chromatin marks H3K9ac and H3K4me3, but not H4K16ac, which is associated with the well-known dosage compensation mechanism via XCU in fruit flies. Whether the histone modifications are a cause or a consequence of XCU in mESC remains an open question. The authors then profiled a series of XX mESC clones harbouring large heterozygous deletions collectively covering the entire X chromosome and showed that (1) no deletion leads to global XCU on the other X chromosome, excluding the existence of a potential “X-upregulation centre” (akin to the “X-inactivation centre” for X-chromosome inactivation); (2) genes within the monosomic region are upregulated; (3) remaining genes in trans are not upregulated. Thanks to the use of both transcriptomic and proteomic approaches, the authors show that upregulation of the monosomic region in the context of the heterozygous deletions, or of the X chromosome in the context of XY and XO cells, happens at both RNA and protein level. Interestingly, and as previously suggested, the authors find that global XCU occurs on a gene-by-gene basis and involves post-transcriptional mechanisms, since for some genes dosage compensation is observed only at the protein level.

Overall, this study is of high quality and provides unexpected results and important insights into how dosage compensation operates in mammals, using a systematic series of chromosomal segment losses along the X chromosome. We would like to recommend this manuscript for publication, however a number of points should be addressed, especially regarding the definition of “upregulated” genes, which has consequences for several of the downstream analyses. While we believe the conclusions will most likely remain unaffected, the analyses might have to be revisited if a new set of “upregulated” genes is defined.

We thank the reviewers for their insightful and constructive comments which have helped improve the manuscript. We have revisited the definition of “upregulated” genes and updated our analyses accordingly, which do not affect the main conclusion of our study.

1/ The authors found that, for genes for which they obtained allele-resolution data from single-cell RNA-sequencing, 40% were significantly upregulated in XO and XY cells compared to XX. Based on Extended Data Fig. 1C, this number seems to be calculated solely on the basis of statistical significance, ignoring the actual fold change. In other words, for some genes the upregulation seems to be very close to zero, yet statistically significant. Do the authors consider that this has biological meaning? Inversely, there are genes for which the fold change is quite considerable but statistical significance is not reached. Maybe the authors could present a more nuanced view with numbers for the different scenarios (40% when considering only statistical significance, X% genes above fold-change bigger than Z, Y% genes when considering both fold-change and statistical significance). How the “upregulated” genes are

defined has an impact on subsequent analyses, so it would make sense to rerun the analyses (at least some) with the alternative set of “upregulated” genes.

We appreciate the helpful suggestions to improve our manuscript. We agree that applying more stringent fold-change cutoffs is important to exclude genes that are statistically significant but exhibit minimal fold changes, thereby focusing our analyses on biologically meaningful gene expression differences. We also agree that presenting a more nuanced view of upregulated genes will strengthen the conclusions of our study.

To address this, we reanalyzed the data using three different fold-change cutoffs, 1.1, 1.2 and 1.5, and reperformed the downstream analyses accordingly. This approach generated new sets of upregulated genes with increasing stringency. Compared to the original 113 upregulated genes (based solely on statistical significance) we identified: 108 upregulated genes using a fold-change cut-off of 1.1, 103 upregulated genes with a cut-off of 1.2, and 64 upregulated genes with a cut-off of 1.5 (see **Rebuttal Figure 1a**).

Rebuttal Figure 1. Identification of upregulated genes with different cut-offs. **a.** Volcano plot of X-linked genes in X-monosomic cells ($X^{\text{Cast}O^{\text{Mus}}}$, $X^{\text{Mus}O^{\text{Cast}}}$, $X^{\text{Mus}Y}$ together) and $X^{\text{Cast}X^{\text{Mus}}}$ cells. P-values are calculated via one-sided Student’s T-test with alternative hypothesis of $X^{\text{monosomy}} > X^{\text{Cast}X^{\text{Mus}}*0.5}$. Horizontal dashed lines represent $P\text{-value} = 0.05$; vertical dashed lines represent fold changes of 1.1, 1.2, and 1.5. $X^{\text{Cast}X^{\text{Mus}}}$ expression values are copy number-corrected (50% of total allelic expression). **b.** Scatter plot of \log_1p expression values of upregulated genes in X^{monosomy} cells by \log fold change between X^{monosomy} and copy number-corrected XX expression. Vertical dashed lines represent fold changes of 1.1, 1.2, and 1.5.

During the reanalysis, we discovered that a small subset of genes ($n=5$) with no expression in XX cells had been included previously, resulting in infinite fold-change values. We excluded these genes from the updated analyses.

We further investigated the relationship between average gene expression and fold change among the upregulated genes. Notably, the five excluded genes between the 1.1 and 1.2 cutoffs were among the highest expressed genes (**Rebuttal Figure 1b**), while genes with lower expression levels tended to exhibit larger fold changes.

We then reran downstream analyses with these updated upregulated gene sets. The 1.1 cutoff set remained significantly enriched for genes encoding complex-forming proteins, but this enrichment was lost at the 1.2 cutoff ($p = 0.05792$) and further diminished at 1.5 ($p = 0.3371$) (**Rebuttal Figure 2a**). None of the upregulated gene sets showed significant enrichment for genes encoding proteins involved in condensate formation or for genes escaping X-chromosome inactivation (**Rebuttal Figures 2b-c**).

Rebuttal Figure 2. Assessing downstream analyses using different upregulated gene sets. a-c. Barplots showing the proportion of genes which code for **a)** complex-forming proteins, **b)** condensate-forming proteins, **c)** escapees in the upregulated (green) and non-upregulated (gray) gene lists for multiple cut-offs (1.1, 1.2, and 1.5). P-values were calculated using the Chi-square test. **d.** Normalized expression in $X^{\text{Cast}}X^{\text{Mus}}$ iPSCs of genes annotated as upregulated (in XO and XY cells, n=103, green) or non-upregulated (n=168, gray). P-value was calculated using a Student's T-test. **e-h.** Violin plots of the **e, g)** mean normalized gene expression, or **f, h)** mean X-to-autosome ratio of upregulated genes in $X^{\text{Cast}}X^{\text{Mus}}$, $X^{\text{Cast}}O^{\text{Mus}}$, $X^{\text{Mus}}O^{\text{Cast}}$, and $X^{\text{Mus}}Y$ cells for the *Mus* (**e, f**) and *Cast* (**g, h**) allele. Dashed line indicates the mean value of XX cells, serving as a reference line for expected expression without upregulation. P-values were calculated via the Tukey HSD test.

When comparing expression levels in XX cells, genes in the 1.1 and 1.2 sets were expressed significantly higher than non-upregulated genes, whereas the 1.5 set was expressed significantly lower (**Rebuttal Figure 2d**). We attribute this to the exclusion of many highly expressed genes by the more stringent 1.5 cutoff (**Rebuttal Figure 1b**).

In response to Comment #4, we also examined the average expression of upregulated genes across cell types for each cutoff. For all cutoffs, the average expression and X:A ratio of upregulated genes was significantly higher in X monosomy cells compared to each allele in XX cells (**Rebuttal Figures 2e-h, Supplementary Fig. 2e-h**).

After careful evaluation, we concluded that a fold-change cutoff of 1.2 strikes the best balance: it excludes genes with minimal upregulation that may lack biological relevance, while retaining genes with incomplete dosage compensation. Importantly, the major conclusions of our study remain unchanged using this cutoff, which also increases confidence in the biological significance of the observed gene expression changes. Specifically, we still detect upregulation for a subset of genes within but not outside large heterozygous deletions (**Figure 4**). Accordingly, we have updated all relevant figures and analyses in the manuscript to use a 1.2 fold-change cutoff for defining transcriptionally upregulated genes (**Figure 1f, h-i, Figure 4g, Supplementary Fig. 2c-n, Supplementary Fig. 3, Supplementary Fig. 4**)

Finally, we generated a gene set defined by the 1.2 fold-change cutoff without applying a statistical significance threshold. The results were consistent with those obtained using both the 1.2 cutoff and significance threshold (**Rebuttal Figure 3**). To maintain rigor, we have chosen to continue using the significance threshold alongside the 1.2 fold-change cutoff when defining upregulated genes in X monosomy.

We appreciate the reviewer's valuable feedback, which has allowed us to refine our analyses and strengthen the robustness of our conclusions.

Rebuttal Figure 3. Omission of significance threshold does not affect findings. **a.** Barplots showing the proportion of genes which code for complex-forming proteins, condensate-forming proteins, or escapees in the upregulated (green) and non-upregulated (gray) gene lists. P-values were calculated using the Chi-square test. **b.** Violin plots of the mean normalized gene expression and mean X-to-autosome ratio from the *Mus* and *Cast* allele of upregulated genes in $X^{Cast}X^{Mus}$, $X^{Cast}O^{Mus}$, $X^{Mus}O^{Cast}$, and $X^{Mus}Y$ cells. Dashed line indicates the mean value of XX cells, serving as a reference line for expected expression without upregulation. P-values were calculated via the Tukey HSD test. **c.** Heatmap showing the fold change in expression of upregulated genes identified from the allelic single-cell analysis between $X^{Monosomy}$ and $X^{Cast}X^{Mus}$. Fold change is calculated between *Cast* expression from either $X^{Cast}O^{Mus}$, Deletion 1, or Deletion 2 cells and total allelic expression divided by 2 from XX cells, or *Mus* expression from $X^{Mus}O^{Cast}$, $X^{Mus}Y$, or Deletion 3 cells and total allelic expression divided by 2 from $X^{Cast}X^{Mus}$ cells.

2/ Why do the authors state that the occurrence of XCU at the protein level is not clear (lines 333-334)? To our knowledge, there is only one study addressing this, which the authors properly cite, and this study concludes that there is compensation at the protein level. Do the authors have reason to believe that the conclusion is disputable? This could be stated if so.

We sincerely thank the reviewer for this important and insightful comment. The study by Wang et al.¹ is indeed a landmark in the field and has greatly informed both our work and the broader

research community. We have no reason to dispute their conclusions. Our intention was to convey that, specifically in mouse pluripotent stem cells with a single active X chromosome, it remains to be directly demonstrated whether X-chromosome upregulation occurs at the protein level. To clarify this, we have revised the manuscript accordingly (page 10 lines 374-375). "XCU has been described to occur at the transcriptional level in mouse embryos and ESCs but its occurrence at the protein level in mouse PSCs remains to be established". We appreciate the reviewer's careful reading and helpful suggestion to improve the clarity of this point.

3/ It would be helpful to acknowledge in the text that for sample X(Mus)O the number of single cells analysed was much lower than for other samples.

We appreciate the suggestion to help improve our manuscript and have made adjustments to the text on page 4 at line 125 to better acknowledge the number of $X^{\text{Mus}}O^{\text{Cast}}$ cells.

4/ Can the authors show the plots in Fig. 1B-C but only for the upregulated genes?

To address this excellent suggestion, we have performed the analysis, which is now shown in **Supplementary Fig. 2e-h**. The average allelic expression of upregulated genes, as well as the average allelic X:A ratio, is significantly higher in cells with X^{monosomy} than in $X^{\text{Cast}}X^{\text{Mus}}$, in line with the conclusion that the X chromosome in XO and XY mouse pluripotent stem cells is upregulated.

5/ The authors state that their findings regarding the histone modifications "overturn the view that these are not enriched on the Xu" (line 202). Could the authors provide a reference for this view? Other studies have associated XCU with specific histone modifications (including a study from the authors).

Indeed a reference for the view that histone modifications are not enriched on the upregulated mammalian X chromosome was missing from the original manuscript. The claim of "no enrichment of active histone marks on the Xu" was most recently made in the paper of Lentini et al.². We have added this reference to the text on page 6 at lines 220-221. We have also included a brief section in the introduction on page 3 at lines 79-88 to address other studies which have investigated histone modifications and XCU^{3,4}.

6/ The authors state that "both species appear to have evolved the use of different histone marks for XCU, H4K16ac in Drosophila, H3K9ac and H3K4me3 in mice". This seems to imply that hPTM are instructive for XCU in mice, which has not really been shown? They might just be a by-product of increased transcriptional activity? Yet, this might not totally be the case either, since the authors show increased hPTM over time even for genes that are not upregulated, suggesting that hPTM can precede increased gene expression; something that the authors do not mention but could be worth discussing. Still, these data would suggest that hPTM are not sufficient to induce XCU, and therefore not instructive (unlike the scenario in Drosophila).

This is an excellent point and we would like to thank the reviewers for their comment. We agree that it remains unknown if hPTMs are instructive for XCU in mice. We have therefore revised the original statement and have made adjustments to the text on page 6 at line 238-

239 to tone down the sentence which now reads: “However, different sets of histone marks appear to be enriched on the X_u in different species, H4K16ac in *Drosophila*, and H3K9ac in mice”.

We also agree it is worth discussing the increase hPTM over time for genes that are not upregulated, suggesting that hPTM can precede increased gene expression, and that hPTM are not sufficient to induce XCU, and are therefore not instructive. We have expanded the discussion in this regard (page 12 at lines 461-464).

7/ The XCU center hypothesis needs to be further and explicitly explained (and would it make sense to call it “Xuc”, analogous to the “Xic”?). This explanation needs to include a definition of the putative “Xuc”, how many would be necessary, how it would work (in cis, in trans) and what would be expected upon its deletion. The schemes provided in Fig. 2B and 2D are not self-explanatory, it is hard to understand what the authors wanted to convey.

We agree that the XCU center hypothesis is an important concept which could be explained in more detail. The manuscript has been edited on page 7 at lines 241-253 to more clearly define the XCU center hypothesis, including a definition, the number of loci required, hypothesized function, and expected results. The schematics depicted in **Figure 1a**, **Figure 2a,b,d**, and **Figure 4a**, have also been reworked in order to be more clear.

8/ The authors use XCU to refer to either what is observed at the level of single genes or at the level of averaged expression levels from an entire chromosome; this sometimes leads to confusion while reading. For example, in lines 243-244, the authors state that “none of the three deletions examined induced XCU across the X in trans”; we suggest that the authors refer to “global XCU” to avoid confusion (the subtlety of “across the X” might be missed), since the deletions do induce XCU in trans (for genes within the deleted region). Likewise, the authors might also use “full XCU” vs “partial XCU” in specific instances. For example, in lines 268-269, the authors state that “none of the 14 new deletions induced XCU for genes located in trans outside of the deleted regions (Figure 2J)” – the authors might consider adding “full” to XCU, given that for some deletions there seems to be some upregulation (deletions C-G and F). Additionally, would it make sense to run statistics on these data?

We thank the reviewer for their insightful and helpful suggestion to clarify the terminology related to XCU in our manuscript. In response, we have replaced the phrase “global induction of XCU across the X-chromosome” with “global XCU” to improve clarity and avoid potential confusion, as suggested. This change has been made on page 7 at lines 269-274.

Furthermore, we have clarified that there was no full XCU for genes located *in trans* outside of the deleted regions, as indicated on page 7 at lines 269. To better distinguish the extent of upregulation, we have introduced and consistently applied the terms global XCU and full XCU throughout the manuscript where appropriate, as suggested (page 4 at line 131, page 7 at line 268-274, page 8 at line 298 and 306).

The scatter plot of mean gene expression per sample (current **Figure 3c**), indeed suggests that several replicates of the deletion samples exhibit slightly altered expression levels compared to the X^{Cast}X^{Mus} controls. However, due to the limited sample size (n=1 or 2), these differences cannot be statistically tested. Therefore, we included an additional plot showing

expression changes for all genes upregulated in $X^{\text{Cast}}O^{\text{Mus}}$ cells (**Figure 3d**). Across all deletion lines, expression levels were consistently and significantly lower than in $X^{\text{Cast}}O^{\text{Mus}}$, indicating the absence of full XCU. Furthermore, none of the deletions, except the AB deletion, resulted in significant expression changes relative to $X^{\text{Cast}}X^{\text{Mus}}$. The AB deletion likely represents an outlier, as deletion of region A or B alone had no significant effect, and the magnitude of change for AB was smaller than in $X^{\text{Cast}}O^{\text{Mus}}$ cells. Collectively, these results do not provide strong statistical evidence for full XCU occurring outside the deleted regions.

9/ When analysing the compensation for genes inside the deletions, in particular Fig. 3E, the authors make no comments or mention to the fact that: (1) for some regions (ITEL and J), there is no XCU (ITEL) or even a decrease in expression levels (J) in XO compared to XX cells; are the changes observed upon deletion related to the data in Fig. 3G? (2) for some deletions (D, K), XCU is not “full” but “partial” (see previous point too). Is this partial XCU for deletions D and K linked to the results shown in Extended Data Fig. 4F? Why do the authors think these deletions are behaving differently?

These are interesting points, and to better understand the compensation levels across deletions, we performed additional analyses. We found that the number of upregulated genes varies between deletion regions (**Supplementary Table 2**). Consequently, when calculating the average expression of all genes within a deletion, the overall upregulation signal can be masked by genes that do not exhibit compensation. To better illustrate the degree of upregulation, we have included new analyses in **Figure 4f**, which presents the expression changes of individual upregulated genes within each deleted region. This gene-level resolution reveals that these genes are consistently upregulated across deletion lines, reaching levels comparable to those seen in $X^{\text{Cast}}O^{\text{Mus}}$ cells.

In conclusion, we do not believe that these deletions behave fundamentally differently from others, which aligns with our model of gene-by-gene regulation. Rather, dosage compensation levels were initially obscured by averaging across heterogeneous gene responses. While we do believe the region-level plots provide an unbiased overview of global XCU, the gene-level analyses are essential for capturing the true extent of compensation. We have updated **Supplementary Table 2**, added **Figure 4f** and revised the main text accordingly (page 8 at lines 316-319). We hope these additional analyses and clarifications address the reviewer’s concerns and improve the interpretability of our findings.

10/ Can the authors show or state an overall quantification of Fig. 3F and 3G? What is the “large majority” (line 289)?

We acknowledge the unclear language originally used in the manuscript, and have now stated the overall quantification and removed the words “large majority”. On page 9 at lines 327-331, the text now reads “We found that 64 of the 65 upregulated genes located within deletions 1-3 were also upregulated in each of the deletions *in trans*”.

11/ The authors asked whether genes upregulated in XO cells were also upregulated when deleted in cells with large heterozygous deletions. Could the authors also do the reciprocal analysis, i.e., looking at whether genes upregulated in the deletions are upregulated in XO? Or is it redundant given a high overlap? This should be mentioned in either case.

Thank you for this interesting suggestion. We have now performed the reciprocal analysis, and the results remain consistent. We identified genes within deletions that are significantly upregulated in cells with the corresponding deletion, and found these genes to also be upregulated in cells with X^{monosomy} . This additional analysis is now included in **Supplementary Fig. 7f**.

12/ Within the deleted regions, how many genes get upregulated? The authors could also provide a table (or equivalent) showing how many genes are contained in each of their deletions (both the ones named with numbers and the ones named with letters).

We agree with the reviewer that this information is helpful, and have included a table with this information in **Supplementary Table 2**.

13/ In Fig. 3F/3G, why are some genes in grey for the XO if the starting point for the analysis is genes that are upregulated in the XO? We thought that only upregulated genes were being shown, but maybe the authors are showing all genes within the deletion? It would be helpful to have the names of the genes below.

Some genes were grey for XO because the average expression of that gene in that particular XO line was 0. The gene is still an “upregulated gene” because the average expression of all X^{monosomy} cells was significantly higher than the average total expression of XX divided by 2 (and the fold change was greater than 1.2). To simplify, we have removed the genes for which the value was grey in the XO line from the plot. Where possible, we have also added gene names to the plot.

14/ For overlapping deletions (e.g., deletion 1 and deletion B), is there concordance in which genes are upregulated?

Due to differences in experimental conditions, including distinct laboratories, sequencing platforms, and cell lines, the most appropriate approach is to compare deletions within the same dataset. Analysis of overlapping deletions within the bulk RNA-seq dataset revealed consistent upregulation of a shared set of genes (**Rebuttal Figure 4**).

Rebuttal Figure 4. Commonly upregulated genes between overlapping deletions. Schematic depiction of the location of three deletions (A, B, and AB) on the X chromosome, with a table denoting the number of (expressed) genes, as well as the number of upregulated genes within each region.

15/ Can the authors add to the scheme in Fig. 2H or 2I how the deletions 1, 2 and 3 overlap with the others? This is represented in Extended Data Fig. 4B, but it would be important to have it in the main figures anyway.

Thank you for this valuable suggestion. We have added the location of deletions 1, 2, and 3 to **Figure 3a**.

16/ In Fig. 3H, the authors should show the results for the parental line of the clones, *Rnf12*^{+/-} *Pdzd4*^{+/+}?

We have reperformed the allelic qPCR and included the parental *Rnf12*^{+/-} line in **Figure 4h**. The expression of *Pdzd4* in the parental *Rnf12*^{+/-} line was comparable to $X^{\text{Cast}}X^{\text{Mus}}$, and the expression in three of the four *Rnf12*^{+/-} *Pdzd4*^{+/-} replicates was higher than both $X^{\text{Cast}}X^{\text{Mus}}$ and the parental *Rnf12*^{+/-} line.

17/ In Extended Data Fig. 2C-E, it is hard to appreciate big differences. Could statistical testing help clarify?

We agree with the reviewer, but unfortunately, direct statistical testing on profile plots is not feasible due to the nature of the data, and such testing is generally not performed. To address this, we quantified the mean signal in the promoter region (1kb region upstream of the TSS, marked by dashed lines), where differences were most pronounced. We then performed statistical testing between the promoter regions of the upregulated and not upregulated genes. The p-values, corrected for multiple testing, are now included in **Supplementary Fig. 3b,d** and **Supplementary Fig. 4b-c,e**. In addition, the mean promoter signal values correspond to the 24-hour time point shown in **Figure 1h-i**, **Supplementary Fig. 3a,c,e**, and **Supplementary Fig. 4a,c,f**. To clarify the connections between the figures, we have expanded the figure legends of all above mentioned figures accordingly.

18/ The authors noted “a slight increase in gene expression in trans in two cell lines, Deletion 1 and 3, but the magnitude of the increase was low” (Lines 244-246). Is this explained by the upregulation observed for genes in trans for the deleted region?

Yes, this is correct. It was our intention to first introduce that the deletions have *some* effect. Then, we look into which genes are driving that effect (inside versus outside). Since the genes outside of the deletions seem to undergo little-to-no change in gene expression, and given that the genes inside the deletion exhibit a large increase in gene expression, we believe that the “slight increase in gene expression *in trans*” is due to the genes inside the deletion.

19/ When discussing the results in the autosomal context (relative to Fig. 4), the authors often refer to genes being “silenced” or “inactivated”, but a more accurate description would be that they see reduced gene expression levels (at the bulk level)? Can we really talk about silenced/inactivated genes without single-cell data?

We agree that the terms “silenced” or “inactivated” could be softened, and we have replaced these expressions with “reduced expression” or “decreased expression” throughout the manuscript (pages 9-10 at lines 353-371).

In response to the question of whether we can discuss reduced/decreased/silenced/inactivated genes without single cell data, we believe that in this context, allele-specific resolution is more critical. Our allele-specific analysis demonstrates that gene expression decreases specifically on the allele expressing *Xist* following doxycycline induction, which strongly supports allele-specific repression. While single cell data can provide additional insights into cellular heterogeneity, the key point here is the allele-specific reduction in expression, which we have robustly measured. To further support this, we have added a new panel in **Supplementary Fig. 8a** illustrating the allele-specific expression changes.

20/ In Fig. 4 (especially C and E), wouldn't it make sense to plot the data only for the genes that are affected by *Xist* expression, instead of the entire 80-160Mb region?

We thank the reviewer for the valuable suggestion. In response, we have generated the requested plot focusing exclusively on genes silenced by *Xist* on chromosome 3, shown in **Rebuttal Figure 5a-b**. When comparing expression of these silenced genes before and after 24 hours of doxycycline induction, we observe a trend toward increased expression from the *Mus* allele; however, this increase does not reach statistical significance.

We propose a biologically relevant hypothesis to explain this observation. *Xist*-mediated silencing is known to be less efficient on autosomes compared to the X chromosome⁵⁻⁷. We hypothesize that highly expressed genes are more resistant to complete silencing by *Xist*. Consequently, dosage compensation would predominantly occur in autosomal genes with lower basal expression levels, resulting in subtler expression changes. This subtlety, combined with the relatively small sample size (n=3), likely limits our ability to detect statistically significant differences. However, when analyzing the entire region, upregulation remains significant (**Figure 5e**).

Overall, these findings are consistent with the notion that autosomal Xist silencing is partial⁸ and gene expression-dependent, which we have now clarified in the manuscript on pages 9-10 at lines 364-371.

Rebuttal Figure 5. No significant increase in expression of genes with reduced expression after addition of dox. a,b. Scatter plots of mean normalized gene expression from the a) *Mus*, and b) *Cast* allele for genes which decreased expression after 24 hours of doxycycline induced *Xist* expression.

21/ How did the authors select the “deletion” lines to analyse by proteomics? In other words, why deletions E and K? Were there specific reasons?

Deletion lines E and K were selected for the proteomics experiment, because these average-sized regions contain the highest proportion of upregulated genes, with consistent and robust expression changes observed across both biological replicates (Supplementary Table 2, Figure 4f). This rationale is now stated in the manuscript on page 10 at lines 378-380 and page 19 at lines 738-742.

22/ A number of sentences need polishing / corrections:

- Lines 86-88 (sentence is not coherent)
- Line 139 (“in cells with XO and XY cells”)
- Lines 144-148 (“which” vs “that”)
- Lines 170-173 (rephrase for simplicity, two sentences could be merged into one)
- Line 257 (assuming the authors mean “syntenic” and not “synthetic”)
- Line 266-267: the results the authors refer to are not shown in Fig. 2J
- Lines 289–290: slightly confusing — the authors mention that the effect is not happening outside of the deletion region but the cited figures only show data for inside the deletion
- Line 347–349: the sentence is unclear and should be simplified
- Extended Figure 6A: typo in y-axis label

We would like to thank the reviewer for their attention to detail and for the effort they put into improving the manuscript. We really appreciate the time and effort. These changes have been made in the manuscript, as well as in the y-axis label of **Supplementary Fig. 9a** (previously **Supplementary Fig. 6a**).

Reviewer #2 (Remarks to the Author):

We would like to thank Reviewer #2 for co-reviewing the manuscript. We appreciate the excellent suggestions that have greatly helped improve our manuscript, and we are happy this could be used as an opportunity to facilitate training in peer review and provide recognition to Early Career Researchers.

Reviewer #3 (Remarks to the Author):

The manuscript addresses the interesting question of whether X chromosome upregulation is controlled via a single regulatory centre (akin to the X inactivation centre), or rather on gene-by-gene and protein-by-protein level. They use pluripotent mouse cells (ES cells and iPS cells) with differing X chromosome complements, along with heterozygous X chromosome fragment deletions in combination with single cell RNA-seq and proteomics to tackle their question. Their data support a gene-by-gene model of X chromosome upregulation that can occur at an RNA level and/or a protein level, frequently both. While upregulation at the RNA level has been reported by several papers, the authors' novel findings relate to tackling the question of whether it is controlled chromosome-wide or not. Interestingly, they also report that upregulation can occur on autosomes (though to a smaller extent), which was previously known. They show it can occur as a function of silencing of one allele rather than only genetic disruption as occurs in cancer. This section is less well explored. Overall, the paper was enjoyable to read, adds to the existing literature on X chromosome upregulation (which has been a controversial topic, so additional well-controlled studies are valuable), and I think will be of interest to the readers of Nature Communications. The major result that dosage compensation is regulated at a gene-by-gene level has major implications for interpretation of disease and potentially also disease treatment and therefore is likely to be relevant beyond the X chromosome regulation field.

There are several areas in the text, figure legends and methods that need clarification and potentially some limited further experiments. My suggested changes aim to provide confidence in the data or to provide clarity for the reasoning and interpretation for readers who aren't entrenched in the X chromosome dosage compensation field.

We thank the reviewer for the thoughtful and encouraging review. We are delighted that the reviewer found our study on gene-by-gene regulation of X-chromosome upregulation interesting and valuable, particularly the novel insights into chromosome-wide control and autosomal upregulation. We appreciate the reviewers' recognition of the importance of our findings for the broader field and their potential implications.

We also thank the reviewer for their constructive suggestions regarding clarifications and additional experiments. We are grateful for the reviewers' positive assessment and helpful feedback, which has strengthened our work. We have carefully addressed these points to enhance the clarity and robustness of our manuscript.

XX, XO and XiXa status of mESC and iPSC systems:

1. It isn't clear why iPSC are used in some sections and ESC in others, and a mix of two are compared. The authors need to provide explanation for their choice, and then also show that there is no difference between the 2 for XCU upregulation purposes, so that the data can be appropriately interpreted. For example, for Figure 1, only the XY samples are ESCs, so it needs to be clear that comparing male to XO and XX iPSC is reasonable. For Figure 2, deletions are made in XX ESC, but there are no control XX ESCs without deletions (presumably the parental line), only iPSC. These controls in each case are essential for appropriate interpretations, particularly when there are very few single cells from XO iPSC in Figure 1 and so there is a greater reliance on XY ESC data.

We thank the reviewer for raising this important point regarding the use of iPSC and ESC lines in our study. We would like to clarify the rationale behind our choice of cell types and provide evidence supporting the equivalence of iPSCs and ESCs for studying X chromosome upregulation.

We used iPSCs primarily because we had well-characterized XX and XO iPSC lines available from a previous study⁹, which had been extensively validated for X chromosome status and dosage compensation analyses. Additionally, we had generated large heterozygous deletions of X-linked fragments in female iPSCs in that study, which were relevant for our current work. We also used ESCs to include important controls, particularly XY ESC lines, which were readily available in our lab. At the start of this project, we did not have access to XY iPSC or XO ESC lines.

With respect to X-chromosome dosage compensation and XCU, our prior work and others' studies have demonstrated that both iPSCs and ESCs behave similarly, female fibroblasts lose XCU upon reprogramming to iPSCs¹⁰, and that XX female ESCs do not exhibit XCU^{2,10}, whereas XO female ESCs and XY ESCs do show XCU². Furthermore, naive mouse XX epiblast cells lack XCU, and XCU is induced upon differentiation, indicating that XCU is an intrinsic and dynamic property of the embryo epiblast and of mouse pluripotent stem cells in vitro^{2,10}. In this context, both iPSCs and ESCs share fundamental functional properties related to X chromosome sensing and dosage compensation by X chromosome upregulation. Our data (this study, Talon et al.)¹⁰ and literature^{2,9} strongly support that XCU occurs similarly in both iPSCs and ESCs, particularly in XY and XO genotypes.

For **Figure 1**, comparing XY ESCs to XX and XO iPSCs is reasonable because the ability to undergo XCU is conserved across these pluripotent stem cell types, and the genotypes are the primary variable influencing XCU status. For **Figure 2**, deletions were made in XX ESCs from the Song *et al.* study⁹. We included iPSC controls because XX ESCs do not exhibit XCU, but XO and XY ESCs do^{2,10}, consistent with our previous findings⁹. Thus, using iPSCs as controls remains appropriate.

In summary, although we used a mix of iPSC and ESC lines due to availability and experimental design constraints, extensive evidence indicates that mouse iPSCs and ESCs behave equivalently with respect to X chromosome sensing and XCU^{2,10}. We believe that this justifies the comparisons made in our study and supports the validity of our interpretations. We have now clarified these points in the revised manuscript to ensure transparency and facilitate accurate interpretation of the data (pages 3-4 at lines 114-116, and page 13 at lines 508-510).

We hope this explanation addresses the reviewer's concerns.

2. Female iPSC and ESC can lose an X chromosome and become XO (which is useful in this study). In serum + LIF culture systems as used here, the cells can also undergo differentiation and become XiXa. The methods and text do not indicate how the XX populations (mESC or iPSC) were checked for their XX vs XO status, and X inactivation status. These details are also critical and need to be included.

We agree that it is important to verify that the cells are not differentiated and to confirm the XX versus XO status of the cell populations, which we have carefully done.

To ensure the cells analyzed were pluripotent and not differentiated, we isolated pluripotent stem cells by flow cytometry prior to Smart-seq3xpress analysis. Specifically, we sorted cells positive for the pluripotency marker SSEA1 to exclude differentiated cells and used only these SSEA1+ cells for sequencing. We have now added **Supplementary Fig. 1a**, which details the gating strategy used for sorting SSEA1+ cells. Furthermore, downstream single-cell RNA-seq analyses confirmed robust expression of pluripotency markers across all cells and lines, supporting that the cells remained undifferentiated (**Supplementary Fig. 5a-b**).

To verify and identify XX versus XO status at the single-cell level, we leveraged the Smart-seq3xpress scRNA-seq data. We plotted cells based on allele-specific expression of *Mus musculus musculus* and *Mus musculus castaneus* reads (**Supplementary Fig. 1b**). This analysis clearly distinguished cells with an XX genotype (expressing both *Mus* and *Cast* reads) from XO cells (expressing either *Cast* reads only or *Mus* reads only) and XY cells (expressing *Mus* reads only). One cell with ambiguous X status was excluded from downstream analyses.

We have now incorporated a detailed description of these procedures in the Methods section (page 17 at lines 612-621), including how pluripotency, differentiation status, and XX versus XO status were assessed. These critical details have been added to enhance transparency and reproducibility.

RNAseq data presentation:

3. The cell-to-cell variability of mean gene expression (summarised for X-linked genes or subsets) is clearly presented in the scRNAseq data. It is also important to see the range of expression across the genes in the plot for the bulk RNA-seq data, by plotting all genes as separate data points, with replicates side by side. This would give a better sense of variability, which is important when sometimes the effects being interpreted are very small e.g. Fig. 1J, Fig. 3E

We thank the reviewer for this excellent suggestion. To better illustrate gene-to-gene variability in the bulk RNA-seq dataset, we have now plotted individual gene expression changes using scatter plots (**Figure 3d, Figure 4f**). Given the wide range of expression values across genes, we calculated the \log_2 -transformed ratio of TPM to the average TPM in XX cells for each upregulated gene, with biological replicates displayed side by side. This visualization clearly demonstrates that the majority of XO-upregulated genes are consistently upregulated across all deletion lines.

We also note that the proportion of upregulated genes varies between deletion regions (**Supplementary Table 2**), which explains why the mean gene expression shown in **Figure 4e** does not always reflect the same degree of upregulation. These new plots provide a more detailed view of variability, which we agree is important (**Figure 3d and Figure 4f**).

XCU hypothesis and histone modifications:

4. The section of 'No evidence for an XCU center' does not lay out the hypothesis that there might be an XCU, akin to a XIC for inactivation. While it is mentioned briefly in the introduction,

a reminder in the first sentence of this section and also relating it to the XIC would help to set the scene for the experiments.

An additional explanation of the XCU center hypothesis would indeed benefit the reader. This comment was addressed in the response to Comment 7 of Reviewer 1 above.

5. The statistical analyses on Figure 1I/J are not immediately easy to read with regards to the timecourse of ChIPseq. Which asterisks refer to which analyses? Can the authors clarify if it was only 0 and 24 hours used for the timecourse analysis, as indicated? If so, why weren't all data points used?

We agree that the statistical analyses shown in **Figure 1h-i** may be challenging to interpret due to the number of comparisons included. In the original version of the figure, we performed three statistical comparisons for each histone mark: (1) between 0 and 24 hours for the upregulated genes (in green), (2) between 0 and 24 hours for the non-upregulated genes (in gray), and (3) between upregulated and non-upregulated genes at 24 hours (on right side).

To improve readability, we have revised **Figure 1h-i**. The updated version now includes only the comparisons between 0 and 24 hours within each gene set (upregulated and non-upregulated). The third comparison (between upregulated and non-upregulated genes at 24 hours) has been moved to the profile plots in **Supplementary Fig. 3** and **Supplementary Fig. 4**, which present the underlying ChIP-seq signal data used for the quantifications in **Figure 1h-i**.

Regarding the time course analysis, we included RNA-seq data from 0, 4, 12, and 24 hours from Zylicz *et al.*, 2019¹¹. Statistical testing was performed only between 0 and 24 hour time points, as these represent the XaXa and XuXi state, respectively. We chose not to include intermediate time points in the statistical analysis because the data do not follow a linear trend, and to our knowledge, there is no appropriate statistical test that robustly captures non-linear dynamics in this context. For example, linear regression would test for a consistent slope, which does not reflect the trends observed in our analyses (see for example **Supplementary Fig. 3c**).

6. While the main data is about the 'active' histone marks, the polycomb marks were analysed similarly, using just the promoter marking. Here an analysis using the enrichment over the whole gene and the upstream region would be better. In addition, can some example plots be included in supp figures of the appearance of different ChIPseq data tracks to allow readers to assess the data for themselves.

We thank the reviewer for these helpful suggestions. In the initial draft, we analyzed all ChIP-seq datasets using the same approach to maintain an unbiased comparison across histone marks. However, we agree that for polycomb-associated marks enrichment over the entire gene and its flanking regions is more informative than promoter-only analyses. We have therefore updated our profile plots to show enrichment across the gene body, including 1.5 kb upstream and downstream (**Supplementary Fig. 3b,d**). This revised visualization captures both promoter and gene body enrichment patterns. Additionally, we have performed gene body quantifications for the polycomb marks (**Supplementary Fig. 3c,e**). These results

suggest that, although these marks change over time, there are no significant gene body differences between upregulated and non-upregulated genes.

We also appreciate the suggestion to provide genome overviews of the ChIP-seq tracks. To address this, we have added several genome browser views to **Supplementary Fig. 4d**, highlighting representative loci of upregulated genes. These examples illustrate increased H3K9ac and H2AK119ub enrichment at promoter regions, consistent with the significant trends observed in our time series analyses (**Figure 1h-i**). In addition, promoter enrichment of H3K27ac and H4ac, both significantly elevated in upregulated genes (**Supplementary Fig. 4c**), is also clearly visible at these loci.

7. Line 170, no enrichment of genes encoding proteins that are in condensates. Given the challenges studying condensates, and the relatively new area of research, a caveat here that not all proteins that undergo condensation may have been identified seems in order.

We agree that this finding may change as the field progresses. We have added a sentence to the results section on page 5 at lines 185-187 to address this caveat.

8. Line 171. The comment that XCI escapees aren't upregulated needs explanation. Readers that aren't XCI aficionados will need an explanation as to why these would have been tested.

We have provided additional explanation regarding our interest in escapees, explaining that escapee genes are expressed from both X chromosomes, and therefore may need compensation in XO or XY cells (page 5 at line 187-189). We hope this improves the clarity of this section for readers who are less versed in XCI.

Deletion interpretations:

9. Line 280/81, Fig 3E. The authors state that similar observations to the 3 large deletions were made with 12 smaller deletions. However, there appear to be differences in some e.g. C, D, I-TEL, J and K. There appears to be no effect in some e.g. C, half the level of upregulation as in monosomy X in others e.g. K, and lack of an effect in the XO in others again (I-TEL, K). So more explanation and discussion is required here, particularly as the results don't easily tally with the data shown in Fig. 3G.

We agree with the reviewer that in this figure (current **Figure 4e**) some deletions do not seem to induce the same degree of dosage compensation as other deletions. We have addressed this comment together with comment #3 above.

In the previous version of the manuscript, **Figure 3g** presented a heatmap of allele-specific upregulation levels. Because of the inherently lower resolution of allele-specific data, the non-allele-specific analysis presented in current **Figure 4f** provides greater power to assess the upregulation status of individual genes across deletions. To avoid potential confusion and to focus on the more robust dataset, we have removed the deletion samples from the allele-specific heatmap in the revised figures (**Supplementary Fig. 7e**). We believe this change improves the clarity of the data presentation and better supports the conclusions drawn regarding dosage compensation.

10. Here it is not clear why these deletions were made in Rnf12^{+/-} cells. This needs explanation. Was there an advantage, or were they just the cells that were around? Was the idea to know which allele had the deletion in some way?

We selected Rnf12^{+/-} mESCs as the starting point because the heterozygous Rnf12 mutation induces full skewing upon XCI, which was relevant for a parallel study. Importantly, RNA-seq analysis revealed no major expression differences between Rnf12^{+/-} and WT cells, supporting their use in this context (**Supplementary Fig. 6c**). A sentence clarifying this rationale has been added to the Methods section (pages 13-14 at line 524-533).

Autosomal Xist interpretation:

11. Fig. 4B is summarised data, but isn't showing downregulation of the genes. Can the authors include a data plot so show the data for silencing?

We have included a plot to better illustrate how the silenced genes were identified (**Supplementary Fig. 8a**). By plotting the fold change in expression after addition of doxycycline, we now show how genes were defined as being downregulated.

12. Here the authors should discuss why the level of upregulation is much smaller than on the X, and also explain their interpretation. On line 321/322 they say the upregulated genes are not higher, in fact they are statistically LOWER in expression than those that aren't upregulated. This should be made clear.

We have rephrased the sentence on page 10 at line 363-365 to better reflect the results, with emphasis on the fact that the expression is significantly lower.

13. They also say there is no significant difference in genes encoding protein complexes, but the difference is larger than those shown for the XCU data but is apparently not significant. This doesn't quite make sense and should be checked. It would be FEWER genes that are part of complexes that are upregulated, contrasting with the XCU data.

We have adjusted the cut-off used to define upregulated genes as a response to Comment 1 from Reviewer 1, which has resulted in the enrichment of genes encoding proteins in complexes no longer being significant ($p=0.05792$). Due to this change, the results from the chromosome 3 data no longer conflict with those found on the X chromosome. We have also double checked, and the difference is indeed not significant ($p=0.08429$). We believe that the difference in scale may also play a role in exaggerating the degree of the effect.

14. These data together suggest that any upregulation observed, which seems smaller in effect size, is perhaps less reliable than the XCU. Could this be because the cells don't tolerate chromosome 3 silencing well, limiting the capacity to see upregulation in trans?

We agree that the degree of dosage compensation by transcriptional upregulation appears lower for the autosomal data than for the X chromosomes. One possible explanation is that cells may not tolerate chromosome 3 silencing well, which could limit the capacity to observe upregulation *in trans*. Since these data derive from another study¹², we cannot confirm whether

increased cell death occurs in these cells. Another possibility is that *Xist*-induced silencing might be less efficient than genetic deletion, potentially affecting the extent of compensation.

Nonetheless, we have clearly observed dosage compensation of autosomal monosomies in the human embryo¹³, indicating that autosomal dosage compensation can occur. Additionally, it is plausible that mammalian cells have evolved more efficient mechanisms to compensate for monosomy of the X chromosome than for autosomes. This would be consistent with the evolutionary degradation of the Y chromosome and the natural occurrence of X monosomy in males, and a single active X chromosome in most female cells, which likely necessitated more robust compensation mechanisms on the X chromosome.

These alternative possibilities warrant further investigation in future studies.

Protein-level compensation:

15. It isn't clear what the overlap is between XY and XO differential proteomic analyses. I also could not find if all samples were ESC here, or a mix of ESC and iPSC. Can the authors include a comparison between XY and XO to give a high confidence list of XCU proteins? If the list is as different as it appears on the Figure, what is the explanation? The labels on Figure 5 of proteins don't allow such a comparison.

We thank the reviewer for pointing this out. All proteomic samples included in this analysis were derived from ESCs. We have now explicitly stated this in the figure legend of **Figure 6** and Methods section (page 18 at lines 704-714).

Regarding the overlap between XY and XO differential proteomic analyses: our current list of XCU proteins (n=45) is based on proteins that are significantly upregulated in both XY vs XX and XO vs XX comparisons. We have now clarified this criterion in the main text (page 10 at lines 396-397) and in **Supplementary Fig. 9c**, which displays a Venn diagram illustrating the overlap between the two comparisons. To further support this, we have added a scatter plot to **Supplementary Fig. 9d**, showing the correlation of upregulation status in XO and XY for all detected proteins. This analysis confirms a positive correlation between upregulated proteins in XY and XO cells and highlights that the high-confidence XCU proteins represent the intersection of both comparisons.

16. Can the authors explain the X:A ratio analysis in Ext Fig 6B – why would XX cells have X:A ratio of >1?

The reviewer raises an interesting point regarding the observed X:A ratios. The X:A ratios were based on proteins detected across all samples. As our initial analysis included only proteins consistently detected in every sample, this introduces a detection bias. X-linked proteins, present in a single genetic copy in the monosomies and typically upregulated but not fully matching autosomal (2n) levels, must be relatively highly expressed to be consistently detected. In contrast, the autosomal protein list, from genes present in two copies, include more low-abundance proteins that are more easily detected in all samples. This is illustrated in a density plot of protein abundances, which shows a larger contribution of lower abundance autosomal proteins compared to X-linked proteins (**Rebuttal Figure 6a**). As a result, the mean

autosomal (A) level used to calculate the X:A ratio is likely underestimated, leading to an artificially elevated X:A ratio in XX cells.

To correct for this detection bias, we recalculated the X:A ratios using an alternative filtering strategy. In addition to requiring protein detection across all samples, we also filtered for proteins with a mean abundance above a defined threshold in XX samples, where both X-linked and autosomal genes are present in two copies. Using this revised method, XX cells show a mean X:A ratio of 1.0, while XY and XO cells show ratios of ~0.74 and 0.69, respectively, consistent with relative X-linked abundances (**Supplementary Fig. 9a**). We have updated **Supplementary Fig. 9b** and the Methods section (page 20 at lines 776-781) with this corrected analysis.

Rebuttal Figure 6. Mean levels in transcriptomics and proteomics datasets.

a. Density plot of protein abundances in XX cells for proteins encoded by autosomal (A, gray) and X-linked (X, green) genes (mean of $n=3$).

b-c. RNA expression (**b**) and protein abundance (**c**) in XX cells (mean of $n=2$ or $n=3$, respectively) for genes with RNA upregulation but no protein-level upregulation (blue) versus those with both RNA and protein upregulation (orange). For each category, the mean level is indicated by a black line.

17. Figure 5F shows 8% of proteins that have RNA upregulation but not protein level. The confidence in this group is presumably low, given proteomics doesn't have the resolution of transcriptomics, and this should be mentioned. It doesn't make a lot of evolutionary sense to have RNA but not protein level upregulation, as potentially a wasted effort in production.

We agree with the reviewer that transcriptional upregulation without a corresponding increase in protein levels is unexpected and may appear biologically inefficient. One likely explanation for this discrepancy is the limited dynamic range and sensitivity of proteomics compared to transcriptomics, which can hinder the confident detection of changes in protein abundance, especially for proteins expressed at lower levels^{14,15}. Indeed, when we examined the expression levels and protein abundances of genes with RNA upregulation in XX cells, we found that those without detectable protein-level changes tend to have lower overall levels (**Rebuttal Figure 6b-c**). Additionally, multiple studies have shown that post-transcriptional regulation, translational control and protein stability can all contribute to RNA-protein

discordance¹⁴, suggesting that these differences may reflect biological meaningful processes beyond our current understanding.

Together, these factors likely underlie the subset of genes showing RNA upregulation without corresponding protein-level changes. We have added a note explaining this point in the main text (page 11 at lines 413-415).

18. It isn't clear why E and K deletions were chosen for proteomic analyses, especially when K doesn't seem to have 'standard' behaviour at the RNA level. Was this the reason why? In addition the XO and XY do not seem to be similar, but this is not explained either by ESC vs iPSC if this is the case, or another reason. This needs to be explained.

Deletion lines E and K were selected for the proteomics experiment, because these average-sized regions contain the highest proportion of upregulated genes, with consistent and robust expression changes observed across both biological replicates (**Supplementary Table 2, Figure 4f**). This rationale is now stated in the manuscript on page 10 at lines 378-380 and page 19 at lines 738-741.

Regarding the observed differences between XO and XY lines: at the RNA level, we see subtle differences in expression between $X^{\text{MusO}^{\text{Cast}}}$ and X^{MusY} cells (**Figure 1b**). We attribute this to a reference bias in mapping between the two cell lines with different genetic backgrounds (*Mus* in $X^{\text{MusO}^{\text{Cast}}}$ is 129, while *Mus* in X^{MusY} is B16), as well as the small sample size of $X^{\text{MusO}^{\text{Cast}}}$ (n=6). This point is discussed in more detail in our response to Comment 1 of Reviewer 4 below. At the protein level, XY samples also show slightly higher upregulation compared to XO samples (**Figure 6**). As all proteomics samples were derived from ESCs, these differences cannot be attributed to ESC versus iPSC state. Instead, they likely reflect intrinsic differences between the specific cell lines used (F1 2-3 for X^{MusY} and F1 2-1 for $X^{\text{CastO}^{\text{Mus}}}$) and the genetic background of the X chromosome in each line (*Mus* versus *Cast*).

19. The description of this final paragraph of the results is fairly repetitive and needs to be improved.

We thank the reviewer for the suggestion, we have removed some of the repetitive text to make this paragraph more digestible (page 11 at lines 420-432).

Methodological details:

20. Detailed methods for culture of the mESC and iPSC on MEF feeder cells are not provided. More details are required.

We have supplemented the methods section "Cell culture" with additional information regarding culture of mESCs and iPSCs on MEF feeder cells.

21. The methods for creation of the deletion lines are hard to follow and additional explanation of why the specific approach was taken would help e.g. why were *Rnf12*^{+/-} cells used as the starting point? How was the neo cassette later deleted? Why was puromycin mentioned?

We acknowledge the comment of the reviewer and have included additional information regarding the generation of the deletion lines on pages 13-14 at lines 501-532.

Small text changes:

22. Line 267, Figure 2J. These data do not show uncorrected expression so don't show that there is higher X-linked expression in XX vs XO cells.

Unfortunately, we referred to the incorrect figure in this sentence. We apologise to the reviewer for the confusion this may have caused and have taken care to refer to the correct figure (**Supplementary Fig. 6f**).

23. Paragraph starting at line 114, refers to results with interpretations embedded prematurely before all data is shown, so should be edited. For example, line 116 says 'As expected, we confirmed the presence of XCU when only one active X chromosome is present,' and similar interpretations throughout. At this stage it is likely more appropriate to say that higher expression was observed in cells with one X rather than 2. Then after Fig 1D/E data is presented with the X:A ratio, it can be interpreted as XCU.

We have adjusted the text to not refer to XCU until after the X:A ratio data has been presented (page 4 at lines 123-134).

24. Line 139 delete 'in cells with'

We have edited this typographical error.

25. Line 190. Native ChIP-seq is stated, but methods mention formaldehyde fixation, so probably native should be removed.

The ChIP-seq dataset we generated for H4K16ac was indeed not native, as it involved formaldehyde fixation as described in the Methods section. The term "native" here refers to the reanalyzed data from Zyllicz et al., 2019¹¹, which were native ChIP-seq experiments.

26. Line 203. The authors claim some active marks are more highly enriched on genes that become upregulated during XCU, but they don't say which these are in the text, which would help interpret the figure.

We thank the reviewer for their suggestion. We have now clarified the text to better describe which marks we are referring to (page 6 at line 222).

Reviewer #4 (Remarks to the Author):

In this study, Allsop and colleagues investigate the mechanisms of gene dosage compensation, particularly X-chromosome upregulation (XCU), using allele- and molecule-resolution transcriptomics in mouse pluripotent stem cells. They show that segmental deletions on one X chromosome can induce gene-specific upregulation in trans, but not global XCU, and that this compensation occurs at both the mRNA and protein levels. The authors further demonstrate that specific active histone modifications (e.g., H3K9ac, H3K4me3) are enriched during XCU. Notably, they also reveal that autosomal dosage compensation can occur via transcriptional upregulation in trans following targeted inactivation in cis, which is a particularly intriguing finding. The data are comprehensive and clearly presented, offering important insights into how mammalian cells sense and respond to gene dosage imbalance through transcriptional compensation. However, several key issues need to be addressed. The specific comments are detailed below.

We sincerely thank the reviewer for their thoughtful and insightful evaluation of our work. We greatly appreciate the positive recognition of our comprehensive data, clear presentation, and the novel insights into gene dosage compensation mechanisms. The reviewer's constructive feedback is invaluable, and we have addressed the specific points raised to further strengthen our manuscript.

Major Comments:

1. Expression level of the Mus allele:

- In Figure 1b, the expression level of the Mus allele in XMusOCast appears lower than in XMusY. Is this difference statistically significant?
- How do the authors interpret this discrepancy?
- A similar concern arises in Figure 3d, can the authors clarify whether this variation is biologically meaningful or due to technical variability?

We thank the reviewer for their concern regarding the difference in expression levels between $X^{\text{MusO}^{\text{Cast}}}$ and X^{MusY} cells. In both **Figure 1b** and **Figure 4d**, the difference in mean allelic X-linked gene expression is statistically significant ($p=3.67e-6$ and $p=1.1e-3$, respectively). We believe this discrepancy arises from the reference bias in mapping between the two cell lines (X^{MusY} is from a B16/Cast background, $X^{\text{MusO}^{\text{Cast}}}$ is from a 129/Cast background), and does not reflect a meaningful biological difference. Additionally, the small sample size of $X^{\text{MusO}^{\text{Cast}}}$ ($n=6$) limits the precision of expression level estimates and reduces statistical power, making it challenging to confidently distinguish technical variation from true biological differences. This technical variability does not impact the conclusions of our study.

- ##### 2. In Figure 1i, the authors report an increase in H2AK119Ub on the active X chromosome (Xa). Given that H2AK119Ub is typically associated with gene repression, how should this observation be interpreted in the context of XCU?

Though we initially also found this to be a confusing result, a recent article has implicated H2AK119ub in both the repression and derepression of genes¹⁶. Therefore, it is possible that H2AK119ub has a potential role in XCU, however we felt this claim would be too far-reaching for our manuscript.

3. Autosomal compensation:

-The autosome-based findings are novel but less deeply explored. Is autosomal compensation a general phenomenon, or limited to specific loci or cell states?

Studies in human cancers and preimplantation human embryos with aneuploidies have shown autosomal compensation, therefore we hypothesize that autosomal compensation is a general phenomenon^{13,17}. Our results suggest that compensation is limited to specific genes. Autosomal dosage compensation in the early human embryo is being explored more thoroughly using single cell multiomics in the study by Gallardo et al.¹³. However it is possible that specific cell states may have an increased propensity towards dosage compensation. For example, it has been reported that B-cells do not lose X-chromosome upregulation upon re-activating their inactive X-chromosome at the RNA level¹⁸, which implies that regulation of dosage compensation is dependent on cell-type.

-In Figure 4e, the upregulation of the Mus allele in the 80–160 Mb region does not appear to reach the level typically observed in canonical XCU. This may be due to insufficient silencing in cis, as suggested by Figure 4c. The authors may consider generating a heterozygous autosomal knockout model to more definitively test whether dosage compensation via transcriptional upregulation in trans can be induced under stronger silencing conditions.

We appreciate the reviewer's thoughtful suggestion. We have attempted to generate heterozygous large autosomal knockouts in mouse embryonic stem cells. However, these cells consistently failed to survive following clonal selection.

4. Mechanistic insight into the sensing process:

While the study shows that cells can detect segmental deletions and compensate in trans, the underlying molecular sensors or signaling pathways responsible for dosage sensing remain unclear. A discussion or hypothesis about what might constitute the sensing mechanism would strengthen the manuscript.

Yes, we expect similar dosage compensation responses in human stem cells and development. Dosage compensation of autosomes has been previously observed in the embryo¹³. We agree with the reviewer, and we are grateful to the reviewer for allowing us to tackle this in the discussion rather than with long and complicated additional experiments. To expand on a potential hypothetical mechanism for dosage sensing, we have added several sentences to the discussion on page 12 at lines 455-461.

5. Whether similar dosage compensation responses are observed or expected in human stem cells or development? The discussion could benefit from a short discussing about this.

We have included a brief section in the discussion to address what could potentially be occurring in humans, as well as the challenges that future studies will encounter when attempting to study dosage compensation in humans (page 12 at lines 468-478).

Minor Comments:

1. Clearly defined "in trans" and "in cis" early in the introduction for general readers.

To assist in making the manuscript more accessible to general readers, we have added definitions of *in trans* and *in cis* to the introduction where applicable (page 2 at line 45 and page 3 at line 102).

2. In Figure 2j, the TPM values for the same cell type vary across different deletion conditions. Did the authors use the same XMusXCast and XMusOCast controls in each of the deletion comparisons?

We thank the reviewer for the question regarding the scatter plot showing the expression levels of genes outside each deletion (current **Figure 3c**). We believe there may be a misunderstanding: the TPM values shown for each deletion region are based exclusively on genes located outside the respective deleted region. Consequently, the gene sets differ between comparisons, which naturally results in variation in TPM values for the same samples. The same $X^{\text{Cast}}X^{\text{Mus}}$ and $X^{\text{Cast}}O^{\text{Mus}}$ control datasets were used across all comparisons, but the plotted values reflect only the subset of genes outside each specific deletion. To clarify this point, we have added an explanatory note to the legend of **Figure 3c** to prevent future confusion.

3. Additionally, in certain cases (e.g., deletion C–G) in Figure 2j, there appears to be a slight induction of XCU. Could the authors provide clarification on whether this represents a meaningful transcriptional compensation or falls within expected noise?

The scatter plot of mean gene expression per sample (current **Figure 3c**) indeed suggests that several replicates of the deletion samples (including deletion C-G) exhibit slightly altered expression levels compared to the $X^{\text{Cast}}X^{\text{Mus}}$ controls. However, due to the limited sample size ($n=1$ or 2), these differences cannot be statistically tested. Therefore, we included an additional plot showing expression changes for all genes upregulated in $X^{\text{Cast}}O^{\text{Mus}}$ cells (**Figure 3d**). Across all deletion lines, expression levels of upregulated genes outside of the deleted regions were consistently and significantly lower than in $X^{\text{Cast}}O^{\text{Mus}}$, indicating the absence of full XCU. Furthermore, none of the deletions, except the AB deletion, resulted in significant expression changes relative to $X^{\text{Cast}}X^{\text{Mus}}$. The AB deletion likely represents an outlier, as deletion of region A or B alone had no significant effect, and the magnitude of change for AB was smaller than in $X^{\text{Cast}}O^{\text{Mus}}$ cells. Collectively, these results do not provide strong statistical evidence for full XCU occurring outside the deleted regions. Therefore, we interpret the modest increases seen in some deletion samples as likely falling within the range of expected experimental variability, rather than representing biologically meaningful transcriptional upregulation.

Reviewer #5 (Remarks to the Author):

We thank you for co-reviewing this manuscript.

References

1. Wang, Z.-Y. *et al.* Transcriptome and translome co-evolution in mammals. *Nature* **588**, 642–647 (2020).
2. Lentini, A. *et al.* Elastic dosage compensation by X-chromosome upregulation. *Nat Commun* **13**, 1854 (2022).
3. Yildirim, E., Sadreyev, R. I., Pinter, S. F. & Lee, J. T. X-chromosome hyperactivation in mammals via nonlinear relationships between chromatin states and transcription. *Nat Struct Mol Biol* **19**, 56–61 (2011).
4. Deng, X. *et al.* Mammalian X upregulation is associated with enhanced transcription initiation, RNA half-life, and MOF-mediated H4K16 acetylation. *Dev Cell* **25**, 55–68 (2013).
5. Tang, Y. A. *et al.* Efficiency of Xist-mediated silencing on autosomes is linked to chromosomal domain organisation. *Epigenetics Chromatin* **3**, 10 (2010).
6. Loda, A. *et al.* Genetic and epigenetic features direct differential efficiency of Xist-mediated silencing at X-chromosomal and autosomal locations. *Nat Commun* **8**, 690 (2017).
7. Pintacuda, G. *et al.* hnRNP-K Recruits PCGF3/5-PRC1 to the Xist RNA B-Repeat to Establish Polycomb-Mediated Chromosomal Silencing. *Mol Cell* **68**, 955–969.e10 (2017).
8. Popova, B. C., Tada, T., Takagi, N., Brockdorff, N. & Nesterova, T. B. Attenuated spread of X-inactivation in an X;autosome translocation. *Proc Natl Acad Sci U S A* **103**, 7706–7711 (2006).
9. Song, J. *et al.* X-Chromosome Dosage Modulates Multiple Molecular and Cellular Properties of Mouse Pluripotent Stem Cells Independently of Global DNA Methylation Levels. *Stem Cell Reports* **12**, 333–350 (2019).
10. Talon, I. *et al.* Enhanced chromatin accessibility contributes to X chromosome dosage compensation in mammals. *Genome Biol* **22**, 302 (2021).
11. Żylicz, J. J. *et al.* The Implication of Early Chromatin Changes in X Chromosome Inactivation. *Cell* **176**, 182–197.e23 (2019).
12. Nesterova, T. B. *et al.* Systematic allelic analysis defines the interplay of key pathways in X chromosome inactivation. *Nat Commun* **10**, 3129 (2019).
13. Fernandez Gallardo, E. *et al.* A multi-omics genome-and-transcriptome single-cell atlas of human preimplantation embryogenesis reveals the cellular and molecular impact of chromosome instability. *bioRxiv* (2023) doi:10.1101/2023.03.08.530586.
14. Liu, Y., Beyer, A. & Aebersold, R. On the Dependency of Cellular Protein Levels on mRNA Abundance. *Cell* **165**, 535–550 (2016).
15. Kumar, D. *et al.* Integrating transcriptome and proteome profiling: Strategies and applications. *Proteomics* **16**, 2533–2544 (2016).
16. Zhao, J. *et al.* H2AK119ub1 differentially fine-tunes gene expression by modulating canonical PRC1- and H1-dependent chromatin compaction. *Mol Cell* **84**, 1191–1205.e7 (2024).
17. Schukken, K. M. & Sheltzer, J. M. Extensive protein dosage compensation in aneuploid human cancers. *Genome Res* **32**, 1254–1270 (2022).
18. Naik, H. C. *et al.* Lineage-specific dynamics of loss of X upregulation during inactive-X reactivation. *Stem Cell Reports* **19**, 1564–1582 (2024).

Response to comments of the reviewers.

Reviewer #1 (Remarks to the Author):

The authors have fully addressed all my comments. This is a very valuable manuscript for the field of X-chromosome regulation, sex chromosomes and dosage compensation.

We thank the reviewer for their kind words, and for their assistance in improving this manuscript.

Reviewer #2 (Remarks to the Author):

We appreciate the contribution to improving and reviewing our manuscript. Thank you.

Reviewer #3 (Remarks to the Author):

The authors have dealt with all my requests and edited the manuscript accordingly. Thank you.

I have just one clarification, related to point 2 of my review, i.e. that there could also be XaXi cells in their culture system. They have done a good job of explaining why the cells aren't differentiated, but that doesn't necessarily mean they didn't become XaXi. Did they check cells did not have Xist as well as two X chromosomes? If yes, that should be included as well.

We would like to thank the reviewer for their work helping us to improve this manuscript. With regards to point 2 of the initial review, we agree with the reviewer. We did not check *Xist* expression by FISH or similar means. However, we believe that the presence of XaXi cells in our XO population would not impact the main conclusions of our paper, as dosage compensation by XCU would still be induced upon inactivation of one of the two X chromosomes in mouse pluripotent stem cells.

Reviewer #4 (Remarks to the Author):

The authors have addressed my comments. Congratulations.

We thank the reviewer for their contribution to our manuscript.

Reviewer #5 (Remarks to the Author):

We appreciate the contribution to improving and reviewing our manuscript. Thank you.